# Glucocorticoid signaling induces transcriptional memory and universally reversible chromatin changes

Melissa Bothe[1], René Buschow[1] , Sebastiaan H Meijsing[1,2]

**Glucocorticoids are stress hormones that elicit cellular responses by binding to the glucocorticoid receptor, a ligand-activated transcription factor. The exposure of cells to this hormone induces widespread changes in the chromatin landscape and gene expression. Previous studies have suggested that some of these changes are reversible whereas others persist even when the hormone is no longer around. However, when we examined chromatin accessibility in human airway epithelial cells after hormone washout, we found that the hormone-induced changes were universally reversed after 1 d. Moreover, priming of cells by a previous exposure to hormone, in general, did not alter the transcriptional response to a subsequent encounter of the same cue except for one gene, *ZBTB16*, that displays transcriptional memory manifesting itself as a more robust transcriptional response upon repeated hormone stimulation. Single-cell analysis revealed that the more robust response is driven by a higher probability of primed cells to activate *ZBTB16* and by a subset of cells that express the gene at levels that are higher than the induction levels observed for naïve cells.**

## Introduction

Transcriptional memory is an adaptive strategy that allows cells to "learn" from a previous transient exposure to an environmental stimulus and orchestrate a more efficient response when the same cue is encountered again. This phenomenon can manifest as a cell's ability to elicit a more robust transcriptional response of signal-inducible genes when these cells were primed by a previous encounter with the stimulus (reviewed in reference 1). Transcriptional memory has been well-studied in plants that have evolved adaptive transcriptional responses to cope with the various environmental stressors they are subjected to and cannot run away from (reviewed in references 2 and 3). It has also been described in other systems, such as responses in yeast to environmental signals (4, 5) and the response of cells of the immune system to cytokines (6). The mechanisms that underly these adaptive strategies include altered binding of poised RNA polymerase II as well as changes in

the chromatin state, particularly persistent changes in histone modifications, chromatin accessibility, and the incorporation of histone variants (1, 2, 3, 4, 7, 8, 9).

The glucocorticoid receptor (GR) is a hormone-inducible transcription factor that regulates the expression of genes involved in diverse processes including development, metabolism and immunity (reviewed in references 10 and 11). Cytoplasmic GR is activated upon the binding of glucocorticoids (GCs), which are secreted from the adrenal cortex in response to various types of stresses including infection, malnutrition and anxiety (12). Glucocorticoids are also released in a circadian and ultradian manner as short, nearly hourly pulses (reviewed in references 13 and 14). Activated GR translocates into the nucleus where it binds to various genomic loci, resulting in the up- or down-regulation of its target genes. Extensive research indicates that GR can function as both an activator and repressor of transcription (15). For transcriptional activation, the paradigm is that direct DNA binding of GR nucleates the assembly of transcription regulatory complexes that modulate target gene expression (15). For transcriptional repression, the proposed mechanisms responsible are often less clear and more diverse. Some studies suggest that GR down-regulates target genes by binding to repressive DNA sequences known as negative glucocorticoid response elements (16, 17). Furthermore, direct DNA binding by GR, for example, to AP-1 response elements, and the recruitment of corepressors such as NCOA2 is linked to the transcriptional repression of associated genes (18, 19, 20). In addition, repression might be mediated by GR tethering to regulatory factors including AP1, NFκB, and NGFI-B. However, several studies challenge the notion that local occupancy is the general driver of transcriptional repression. Instead, these studies argue that transcriptional repression might be driven by the redistribution of the binding of other transcription factors and coregulators and by alternation of the chromatin structure at enhancers that are not directly occupied by GR (21, 22, 23, 24).

Genomic GR binding is associated with a number of changes to the chromatin state (25). For instance, GR binding induces changes in histone modifications by recruiting cofactors, such as histone methyltransferases and acetyltransferases, that induce post-translational modifications of proteins including histones (15, 26, 27, 28). Moreover, GR binding is associated with chromatin remodeling resulting in local

---

[1]Max Planck Institute for Molecular Genetics, Berlin, Germany   [2]Max Planck Unit for the Science of Pathogens, Berlin, Germany

Correspondence: meijsing@mpusp.mpg.de

increases in chromatin accessibility at its binding sites (29, 30, 31, 32). GR activation also induces chromatin decompaction at a scale that is detectable by light microscopy (33) and stabilizes long-range chromatin interactions as shown by high-throughput sequencing-based methods such as Hi-C (31, 34, 35).

Interestingly, several studies indicate that GR-induced chromatin changes can persist after the withdrawal of hormone. For example, a genome-wide analysis showed that GR-induced increases in chromatin accessibility are maintained for at least 40 min after hormone withdrawal at a subset of GR-binding sites (31). Another study using a genomically integrated mouse mammary tumor virus-fragment showed that changes may even persist much longer with GR-induced increases in DNase I-sensitivity of this locus persisting for more than 20 cell divisions after hormone withdrawal (36). Similarly, large-scale chromatin unfolding that occurred at the *FKBP5* locus upon GR activation persisted for up to 5 d after hormone washout (33), suggesting that GR binding can induce a long-lived chromatin-based "memory" of GR binding. The persistent GR-induced chromatin changes could result in a different transcriptional response of genes when cells are exposed to hormone again. However, if this is indeed the case is largely unknown.

In this study, we sought to (1) link changes in chromatin accessibility upon GC treatment to gene regulation, (2) investigate whether long-lived chromatin changes are a commonly observed feature of GR activation, and (3) whether a previous exposure to GCs can be "remembered" and result in an altered transcriptional response upon a second GC exposure. Our data uncover global increases and decreases in chromatin accessibility that coincide with increased and decreased gene expression, respectively. Even though we find that the changes in chromatin accessibility are universally reversible, we also find indications that cells may "remember" a previous exposure to hormone in a gene-specific and cell type-specific manner.

## Results

### Linking GR occupancy, chromatin accessibility, and gene regulation

To better understand the link between GR binding and changes in chromatin accessibility, we mapped genome-wide changes in chromatin accessibility upon GC treatment in A549 cells by assay for transposase-accessible chromatin using sequencing (ATAC-seq). Specifically, we analyzed changes that occur after a 4- or 20-h treatment with either dexamethasone (Dex), a synthetic GC, or with the natural hormone hydrocortisone (Cort). In agreement with previous studies (29, 30, 31, 32), many sites showed an increase in chromatin accessibility ("opening sites" for both hormone treatments when compared with the control treatment [Figs 1A and B and S1A]). Interestingly, chromatin accessibility was reduced for an even larger number of sites ("Closing sites," Figs 1A and B and S1A), corroborating findings in macrophages that also reported both increases and decreases in chromatin accessibility upon GC treatment (21). Correlation analysis between Dex- and Cort-treated

ATAC-seq samples revealed a high Pearson correlation coefficient (Fig S1B), justifying our use of the two hormone treatments as biological replicates. We could validate several examples of opening and closing sites and noticed that opening sites are often GR-occupied whereas closing sites are often not occupied by GR (Fig 1B and C). For a systematic analysis of the link between GR occupancy and chromatin accessibility changes, we integrated the ATAC-seq results with available GR ChIP-seq data (37). This analysis showed that GR binding is observed at most opening sites (Figs 1A and S1D). In contrast, only a minor subset of closing sites shows GR ChIP-seq signal (Figs 1A and S1D). In line with these observations, motif enrichment analysis revealed that the GR consensus recognition motif was enriched at opening yet not at closing sites (Fig S1C) indicating that most closing sites lack direct DNA binding by GR. Similarly, H3K27ac levels, a mark of active chromatin, increase at opening sites whereas levels decrease at closing sites upon hormone treatment (Fig 1A). In addition, because H3K27ac levels appear to show a modest decrease at sites of non-changing accessibility one could speculate that GR activation induces a global redistribution of H3K27ac (Fig 1A). These findings are indicative of a redistribution of the enzymes (e.g., p300, [38]) that deposit the H3K27ac mark upon GC treatment, which has been described in a mouse mammary epithelial cells (39 *Preprint*) and for the estrogen receptor, a GR paralog (40). To test the effect of GC treatment on global p300 binding in A549 cells, we intersected our data with published p300 ChIP-seq data (41). This analysis showed a marked increase in p300 signal at GR-occupied opening sites whereas p300 is lost from closing sites that are typically not GR-occupied (Fig S1E).

Next, we set out to assess the link between changes in chromatin accessibility, GR binding and gene regulation. Therefore, we generated total RNA-seq data and analyzed read coverage at introns as a proxy for nascent transcript to capture acute transcriptional responses (42) and defined three categories of genes: up-regulated (295), down-regulated, (110) and nonregulated (randomly sampled 500). For each gene category, we scanned a window of 100 kb centered on the transcription start site (TSS) of each gene for opening, closing, and non-changing ATAC-seq peaks, whereas removing sites of increasing accessibility that overlapped TSSs of up-regulated genes and sites of decreasing accessibility that overlapped TSSs of down-regulated genes. Consistent with expectation, opening peaks and GR peaks are enriched near up-regulated genes (Fig 1D). Moreover, multiple opening sites and GR peaks are often found near up-regulated genes (Fig S1G). Conversely, down-regulated genes are enriched for closing peaks. GR peaks are also enriched near repressed genes (Figs 1D and S1F) indicating that for some genes, repression might be occupancy driven. However, the enrichment is markedly lower than for upregulated genes and repressed genes are not enriched for genes with multiple GR peaks (Fig S1G). Similarly, analysis of a U2OS cell line stably expressing GR (U2OS-GR, [43]) showed a modest enrichment of GR peaks near downregulated genes (Fig S2D) and no clear enrichment of repressed genes with multiple GR peaks (Fig S2E). Furthermore, closing peaks, which show GC-induced loss of H3K27ac levels, mostly lack GR occupancy and enrichment of the consensus recognition motif (Fig S2A–C), were enriched near repressed genes.

Taken together, our results further support a model put forward by others (21, 22, 39 *Preprint*, 44) in which transcriptional activation

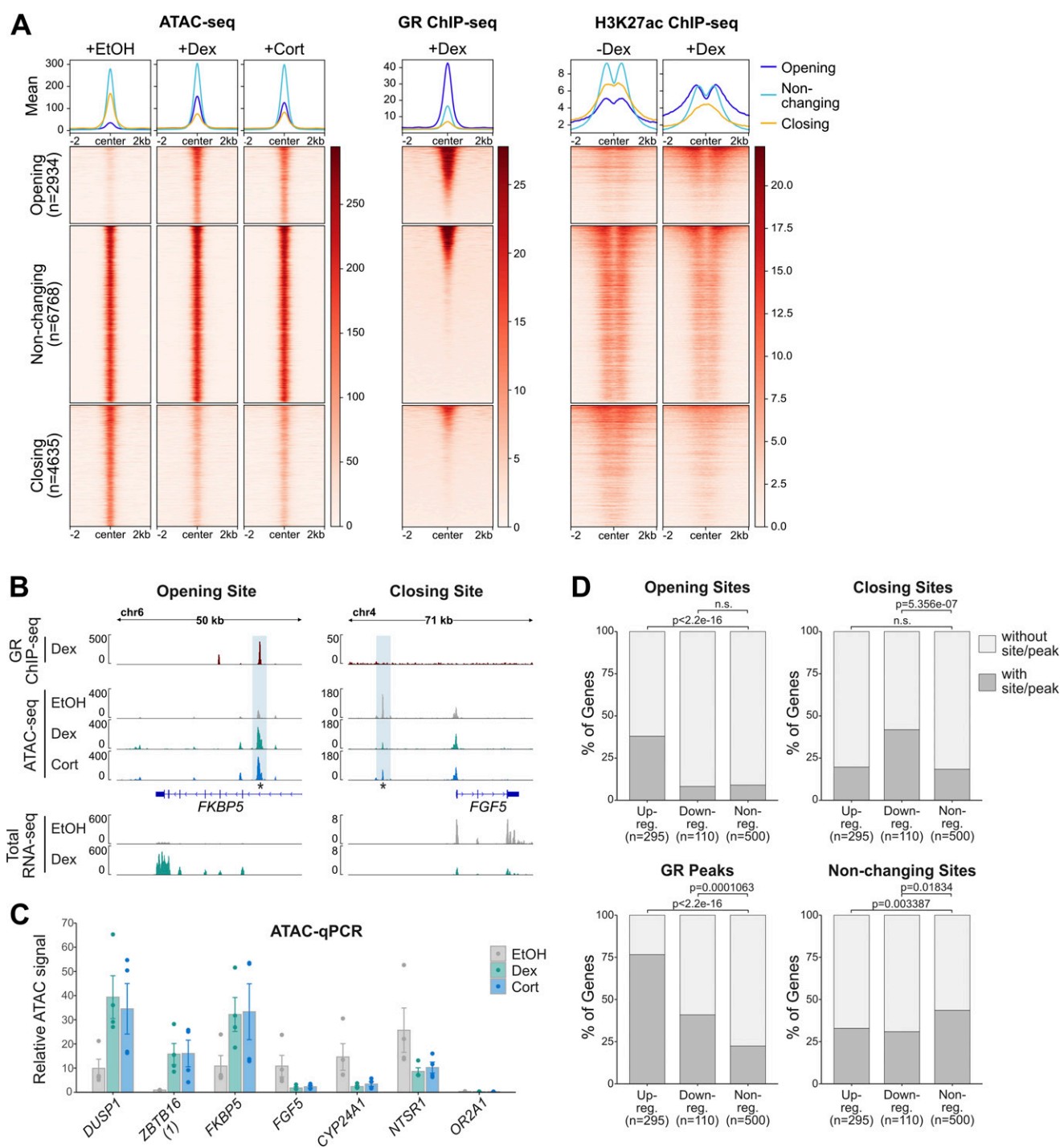

**Figure 1. Integrated analysis of genome-wide changes in chromatin accessibility and transcript levels upon glucocorticoid receptor (GR) activation.**
**(A)** Heat map visualization and mean signal plot of ATAC-seq (normalized), GR ChIP-seq (RPKM-normalized; 100 nM Dex, 3 h; data from reference 37), and H3K27ac ChIP-seq (RPKM-normalized; ±100 nM Dex, 4 h; data from reference 41) read coverage in A549 cells at sites of increasing ("opening"), non-changing ("non-changing"), and decreasing ("closing") chromatin accessibility upon hormone treatment (±2 kb around center). Heat maps are sorted by GR ChIP-seq signal in descending order. For ATAC-seq, cells were treated with EtOH vehicle (20 h), Dex (100 nM, 20 h) or Cort (100 nM, 20 h). **(B)** Genome browser visualizations of the *FKBP5* and *FGF5* loci in A549 cells showing GR ChIP-seq (100 nM Dex, 3 h; RPKM-normalized; data from reference 37), ATAC-seq (normalized) and RNA-seq (RPKM-normalized, merge of three replicates) signal tracks. For ATAC-seq experiments, cells were treated with EtOH (20 h), Dex (100 nM, 20 h) or Cort (100 nM, 20 h). For RNA-seq, cells were treated as detailed in Fig 3A, receiving a vehicle control treatment (4 h), followed by a 24-h hormone-free period and a subsequent 4 h Dex- (100 nM) or EtOH-treatment ("−−": EtOH, "−+": Dex). Opening/closing site is highlighted with blue shading. The asterisk marks the position of the qPCR primers for the analysis shown in (C). **(C)** ATAC-qPCR at sites opening or closing upon hormone treatment near indicated genes in A549 cells. Cells were treated with EtOH (20 h), Dex (100 nM, 20 h), or Cort (100 nM, 20 h). Mean ATAC signal (normalized to gDNA) ± SEM (n = 4) is shown. **(D)** Stacked bar graphs showing the percentage of genes in A549 cells of each category (up-regulated, down-regulated, and nonregulated) that have at least one peak/site for each type (opening, closing, and non-changing sites and GR peaks) within ±50 kb around the transcription start site. GR peaks represent the intersect of peaks called in both replicates (data from reference 37). *P*-values were calculated using a Fisher's exact test. NS, not significant.

by GR is driven by local occupancy whereas transcriptional repression, in general, does not require nearby GR binding. Instead, our data are consistent a "squelching model" whereby repression is driven by a redistribution of cofactors away from enhancers near repressed genes that become less accessible upon GC treatment yet lack GR occupancy. However repression may also be driven by other mechanisms.

## GR-induced changes in chromatin accessibility are universally reversible

A previous study reported that a subset of opening sites remains open 40 min after hormone withdrawal, indicating that cells retain a "memory" of previous hormone exposure (31). Memory may even be "long term," given that a locus-specific study with a genomically integrated mouse mammary tumor virus showed GC-induced opening that persisted for more than 9 d (36). Here, we set out to test if long-term memory of opening sites is a general phenomenon and to expand the analysis to loci with reduced chromatin accessibility upon GC treatment. To mirror the study that reported persistent changes after more than 9 d (36), we treated A549 cells for 20 h with either Dex, Cort, or EtOH as vehicle control followed by a hormone-free washout period (Fig 2A). However, instead of 9 d, we decided to assay chromatin accessibility by ATAC-seq after a 24-h washout period, which we reasoned would be more likely to reveal persistent changes. Analysis of the ATAC-seq data showed that the vast majority of opening sites revert to their untreated chromatin accessibility levels (2,879/2,934 opening sites, Fig 2B). Similarly, the vast majority of closing sites are transiently closed upon hormone exposure (4,593/4,635 closing sites, Fig 2B). Correlation analyses revealed high Pearson correlation coefficients between hormone-treated samples after washout as well as between vehicle-treated samples before and after washout (Fig S3A), showing concordance between replicates. To test if the small number of opening sites with apparent persistent increases in chromatin accessibility were true or false positives of our genome-wide analysis, we performed ATAC followed by qPCR on sites which exhibited the most pronounced residual accessibility (example of a candidate site with persistent opening shown in Fig 2C). The ATAC-qPCR results, however, did not validate the existence of maintained accessibility for these sites and any differences in ATAC signal between basal levels and following hormone washout were, if present at all, subtle (Fig 2D). To test if persistent GC-induced changes in chromatin accessibility can be observed in another cell type, we performed ATAC-seq after Dex or EtOH and subsequent washout in U2OS-GR cells. Consistent with our findings in A549 cells, we found that the vast majority of both opening and closing sites revert to their initial state after hormone withdrawal (Fig S3B and C). Correlation analysis further revealed concordance between the vehicle-treated samples before and after washout (Fig S3D). However, in contrast to our findings in A549 cells, we could validate maintained accessibility at candidate loci by ATAC-qPCR in U2OS-GR cells (Fig S3E). In addition, H3K27ac ChIP experiments showed that GC-induced increases in H3K27ac levels were maintained 24 h after hormone withdrawal at persistent opening sites whereas they reversed to their basal level for sites with transient opening (Fig S3E). To determine whether GR still occupies persistent opening sites, we performed GR ChIP and ChIP-seq on cells 24 h after hormone withdrawal. As expected, we found that GR occupancy was lost at sites with transient opening (Fig S3E and F). In contrast, GR still occupied persistent opening sites 24 h after hormone withdrawal (Fig

S3E and F) indicating that the sustained accessibility is likely the result of residual GR binding to those sites. This could be linked to the fact that the U2OS cells we used in our study overexpress GR and accordingly have a higher number of GR binding sites and opening sites than A549 cells. Another explanation for the sustained occupancy at persistent loci is that a small fraction of GR is still hormone-occupied despite the 24-h washout and dissociation half-life of Dex of ~10 min (45). Interestingly, at low (sub-$K_d$), Dex concentrations, GR selectively occupies a small subset of the genomic loci that are bound when hormone is present at saturating concentrations (46). To test if GR preferably binds at persistent sites at low hormone concentrations, we performed GR ChIPs at different Dex doses (Fig S3G) and found residual occupancy at persistent opening sites at very low Dex concentrations (0.1 nM), whereas GR occupancy was no longer observed at sites with transient opening (Fig S3G). This suggests that the few sites with persistent opening are likely a simple consequence of an incomplete hormone washout and associated residual GR binding.

Together, we conclude that GC-induced changes in chromatin accessibility are universally reversible upon hormone withdrawal with scarce signs of "long-term" memory of previous hormone exposure.

## Prior exposure to GCs results in a more robust regulation of the GR-target gene *ZBTB16*

Next, we were interested to determine if a previous exposure to GCs influences the transcriptional response to a second exposure of the same cue. Therefore, we compared the transcriptional response to a 4-h Dex treatment between "naïve" cells, which have not been exposed to a prior high dose of hormone, and cells that were "primed" by a prior 4-h Dex treatment followed by a 24-h-hormone-free recovery phase (Fig 3A). To capture acute transcriptional responses, we performed total RNA-seq and analyzed read coverage at introns to capture nascent transcripts (42). As expected, we found that GC treatment of naïve cells resulted in the up- or down-regulation of many transcripts (Fig 3B). Importantly, comparison of the basal expression levels between primed cells and naïve cells showed that nascent transcript levels did not show significant changes for any gene indicating that the transcriptional changes were universally reversed after the 24-h hormone-free washout (Fig 3C). When comparing the transcript levels for Dex-treated cells between naïve and primed cells, we found that transcript levels were essentially the same with a single significant exception, the GR-target gene *ZBTB16* (zinc finger and BTB domain containing gene 16, Fig 3D and E). To further substantiate this finding, we performed qPCR which confirmed that a prior hormone treatment resulted in a more robust up-regulation of *ZBTB16* mRNA levels whereas prior treatment did not change the up-regulation of two other GR target genes, *GILZ* and *FKBP5* (Fig 4). Notably, increased mRNA levels are not a simple consequence of mRNA accumulation given that mRNA levels after washout (+- treatment condition) return to the levels observed for untreated naïve cells (Figs 3E and 4A). A more robust *ZBTB16* activation upon GC treatment was also observed when cells were primed followed by a longer, 48-h-hormone-free recovery phase (Fig S4A), suggesting that cells "remember" a prior hormone exposure through a cell division cycle. Moreover, *ZBTB16* up-regulation was further enhanced when cells were exposed to GCs a third time (Fig 4B). To test if priming of the *ZBTB16* gene is cell

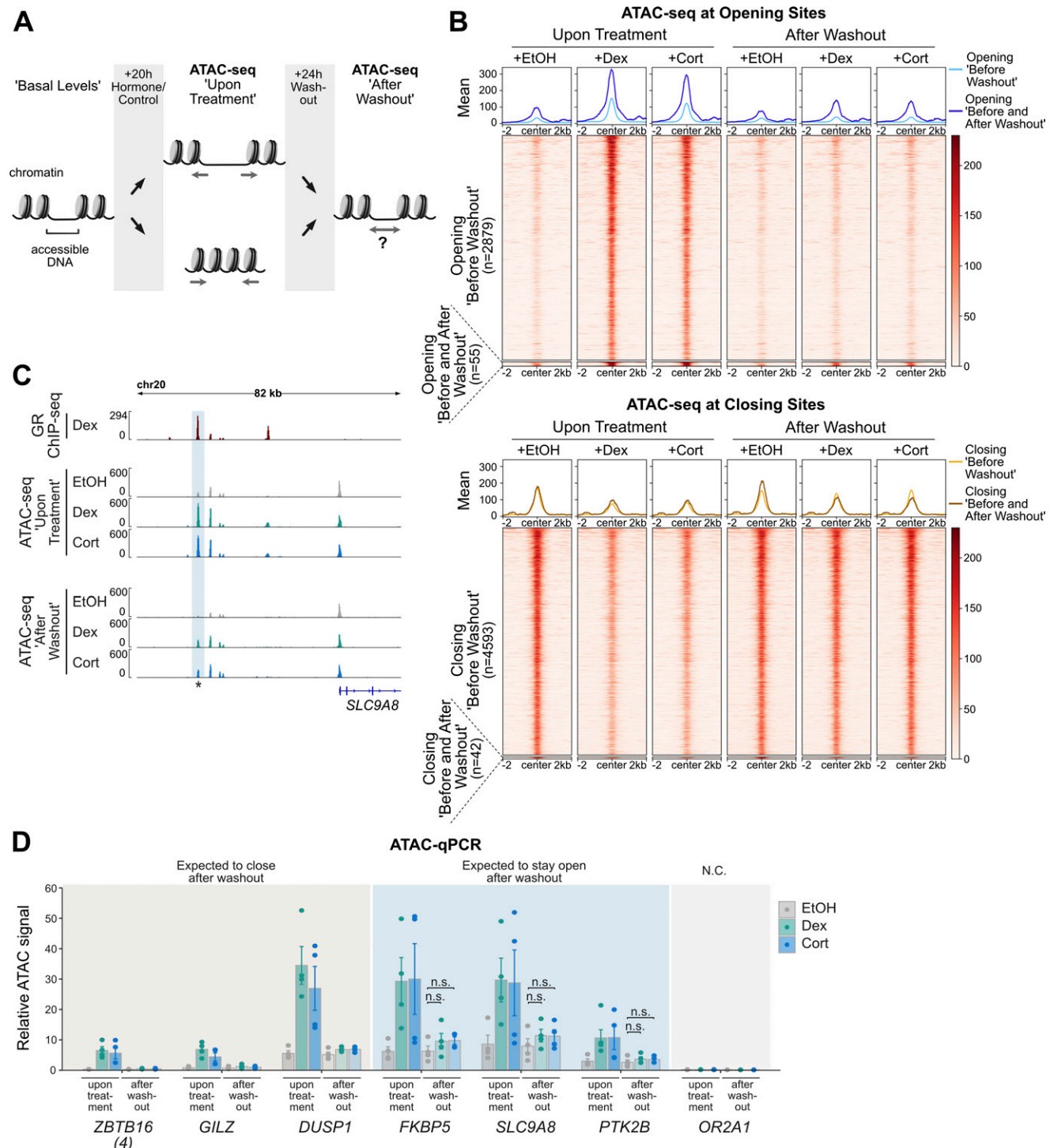

**Figure 2. GC-induced changes in chromatin accessibility are universally reversible.**
**(A)** Cartoon depiction of the experimental design to assess changes in chromatin accessibility upon hormone treatment and after hormone washout. ATAC-seq was performed (1) on A549 cells treated with Dex/Cort (100 nM) or EtOH for 20 h ("upon hormone treatment") and (2) on cells treated with Dex/Cort (100 nM) or EtOH for 20 h, followed by hormone washout and incubation in hormone-free medium for 24 h ("after washout"). **(B)** Heat map visualization and mean signal plot of normalized ATAC-seq read coverage at opening sites (top) and closing sites (bottom) (±2 kb around center). Heat maps are sorted by GR ChIP-seq signal as in Fig 1A. Cells were treated as described in (A). The regions are divided into sites which show reversible increased/decreased accessibility upon hormone treatment and regions with persistent changes (Opening "Before and After Washout"). **(C)** Genome browser visualization of the *SLC9A8* locus in A549 cells showing GR ChIP-seq (100 nM Dex, 3 h; RPKM normalized; data from reference 37) and ATAC-seq (normalized) signal tracks. For ATAC-seq, cells were treated as described in (A). Candidate site with persistent increased accessibility after hormone washout is highlighted. The asterisk marks the position of the qPCR primers for the analysis shown in (D). **(D)** ATAC-qPCR of sites opening upon hormone treatment near indicated genes in A549 cells. Cells were treated as described in (A). Regions which are expected (based on the ATAC-seq data) to close or remain accessible after hormone washout are indicated. Mean ATAC signal (normalized to gDNA) ± SEM (n = 4) is shown. N.C. Negative Control. *P*-values were calculated using a two-tailed *t* test. n.s., not significant.

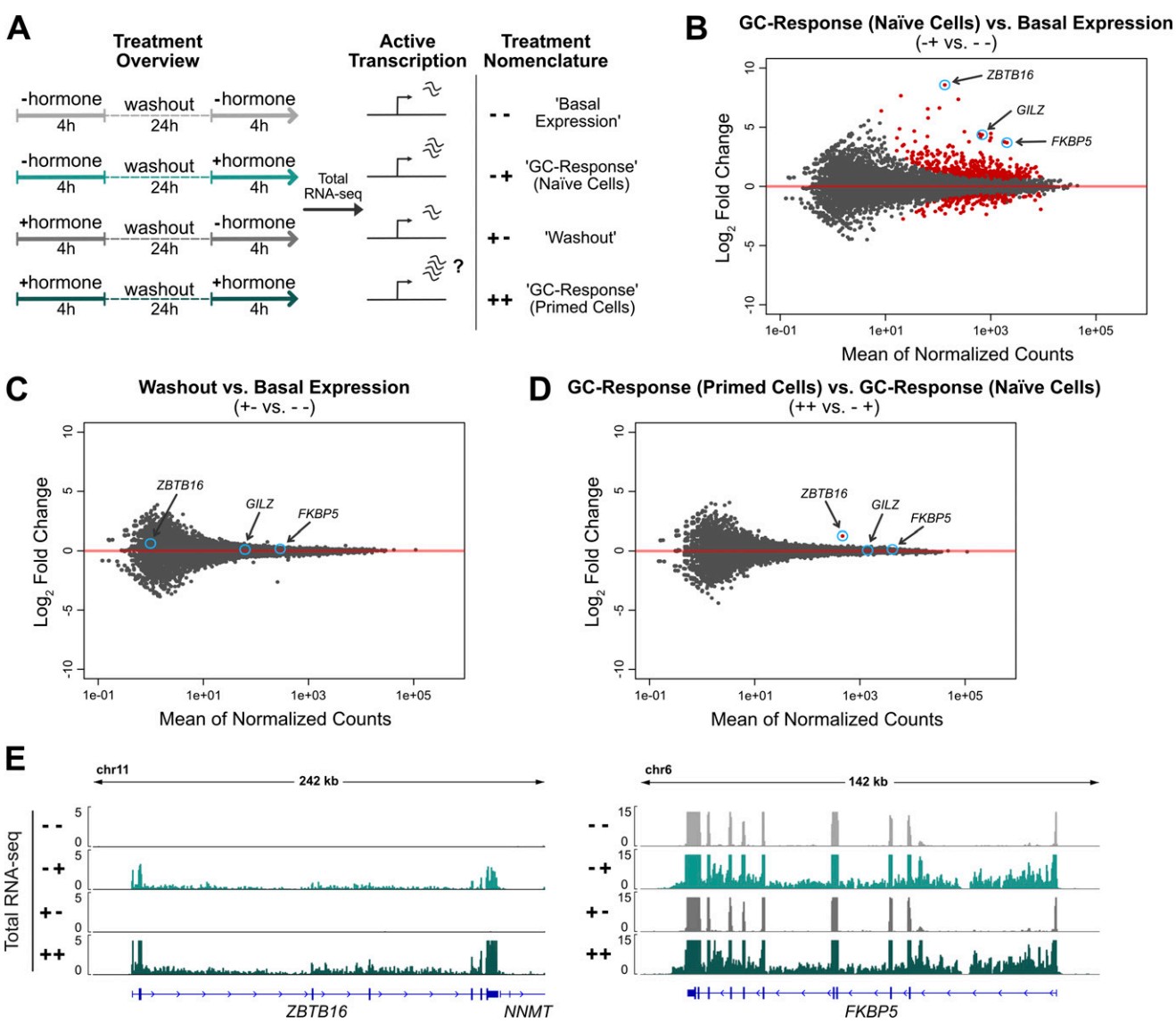

**Figure 3. Priming results in a more robust transcriptional response of the *ZBTB16* gene upon repeated GC exposure.**
**(A)** Cartoon illustrating our experimental set-up to study how priming by a previous hormone treatment influences a subsequent transcriptional response to hormone. Cells were initially treated with 100 nM Dex ("primed") or EtOH (naïve) for 4 h. Subsequently, hormone was washed out and cells were cultured in hormone-free medium for 24 h, after which cells were treated again with either Dex (100 nM) or EtOH for 4 h. **(B, C, D)** MA-plots illustrating the mean of normalized counts and the log2 fold changes for genes when comparing cells subjected to the following treatments: (B) "– +" versus "– –," (C) "+ –" versus "– –," (D) "++" versus "– +". Differential expression analysis was carried using total RNA-seq data and quantifying read coverage within introns. The cells were treated as described in (A). Red dots represent genes significantly up- or down-regulated (FDR < 0.001). **(E)** Genome browser visualizations of the *ZBTB16* and *FKBP5* loci in A549 cells showing total RNA-seq (RPKM-normalized, merge of three replicates) signal tracks. The cells were treated as described in (A).

type-specific, we analyzed how priming influences a subsequent response to hormone in U2OS-GR cells. When we used Dex to prime U2OS-GR cells, we observed that basal *ZBTB16* levels remained high after hormone washout consistent with residual GR occupancy (Fig S4C). Therefore, we now used Cort, which has a shorter dissociation half-life than Dex (45), to prime cells. Consistent with our observations using Dex, we found that priming A549 cells with Cort resulted in a more robust up-regulation of the *ZBTB16* gene upon a subsequent hormone exposure (Fig S4B). In contrast, priming of U2OS-GR cells did not alter the response of *ZBTB16* to a subsequent hormone exposure (Fig S4D).

Together, our results indicate that a previous exposure to GCs does not result in large-scale reprogramming of the transcriptional responses to a second exposure, yet argue for cell type-specific transcriptional memory for the *ZBTB16* gene.

### Short-term maintenance of GC-induced changes in chromatin accessibility, 3D genome organization, and PolII occupancy

To determine if persistent changes in the chromatin state contribute to transcriptional memory of the *ZBTB16* gene, we investigated changes in (1) H3K4me3 and H3K27me3 levels, (2) chromatin

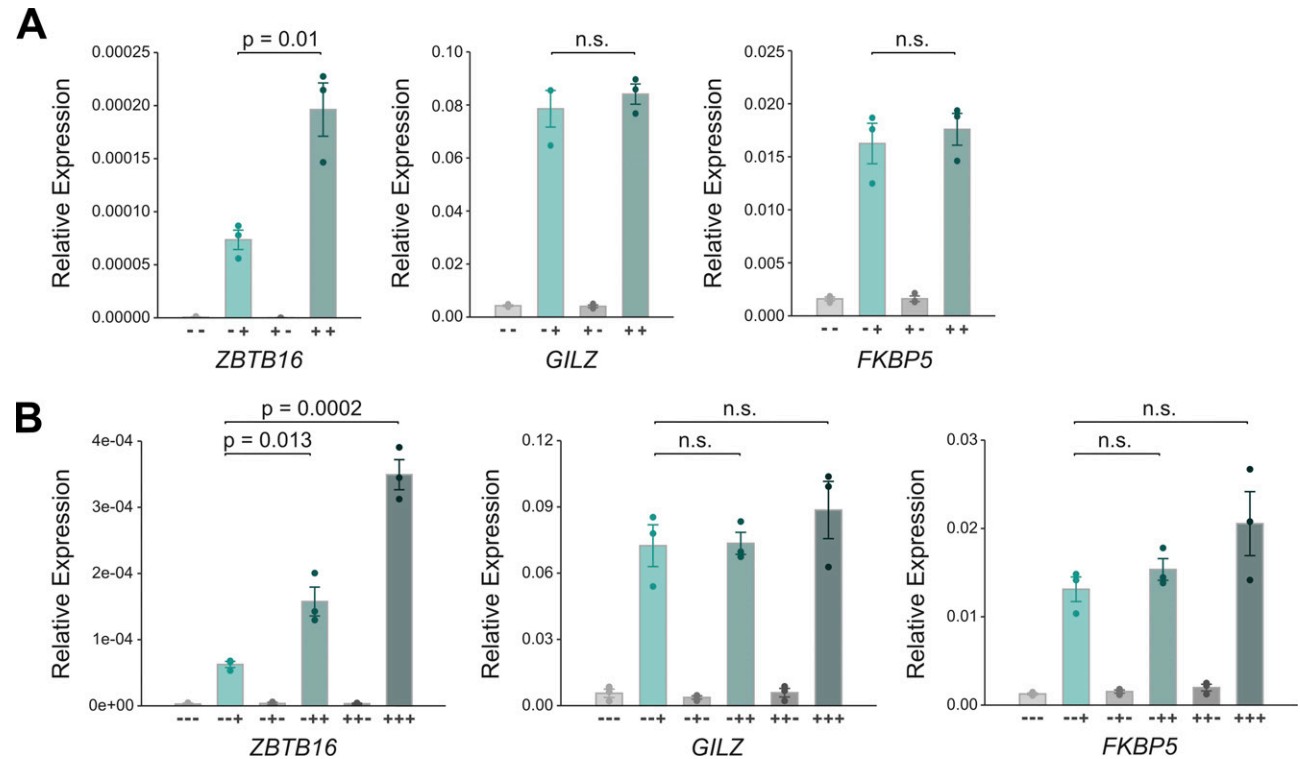

**Figure 4.   Validation that priming results in a more robust transcriptional response of the ZBTB16 gene.**
**(A)** RT-qPCR results for GR-target genes in A549 cells. The cells were treated as detailed in (Fig 3A). Mean expression relative to *RPL19* ± SEM (n = 3) is shown. *P*-values were calculated using a two-tailed *t* test. n.s., not significant. **(B)** RT-qPCR results for glucocorticoid receptor target genes in A549 cells. Cells were treated similarly as detailed in (Fig 3A), except that they received three rounds of hormone treatment: after the second treatment, cells were subjected to washes and cultured in hormone-free medium, and treated again 48 h after the first washout with either Dex (100 nM) or EtOH for 4 h. Mean relative expression to *RPL19* ± SEM (n = 3) is shown. *P*-values were calculated using a two-tailed *t* test. n.s., not significant.

accessibility, (3) RNA polymerase II (PolII) occupancy at the promoter, and (4) 3D chromatin organization. Analysis of available ChIP-seq data for A549 cells (41) showed that *ZBTB16* is situated within a repressed genomic region, with high H3K27me3 levels and low or absent H3K4me3 and H3K27ac signal, respectively (Fig 5A). Among genes up-regulated upon hormone treatment, the presence of H3K27me3 is not unique to *ZBTB16* (Fig 5B). Moreover, ChIP experiments we performed showed that priming did not result in significant changes in H3K4me3 or H3K27me3 levels for either treated cells or after hormone washout indicating that these marks likely do not contribute to the transcriptional memory observed (Fig 5C). Similarly, increases in chromatin accessibility at GR-occupied loci that occur near the *ZBTB16* gene reverted to their uninduced levels after hormone washout (Fig 5D and E). Furthermore, although priming resulted in the accumulation of PolII and PolII phosphorylated at serine 5 at the promoter of *ZBTB16*, we did not detect any perceptible maintenance of this accumulation after washout (Fig 5F). Next, we decided to investigate a possible role of 3D chromatin organization, given that GR activation results in chromatin decompaction that persisted for 5 d (33). Moreover, a recent study into the mechanisms responsible for priming by interferons indicated a role for cohesin and topologically associating domain in transcriptional memory (6). To probe for changes in 3D organization, we performed circular chromosome conformation capture (4C) experiments (47) using either the promoter of the *ZBTB16* gene or an

intronic GR-occupied region as viewpoint. The 4C-seq experiments showed an increase in the relative contact frequency between the *ZBTB16* promoter and a cluster of intronic GR-binding sites upon hormone treatment (Fig 5G). However, after washout this increased contact frequency was reversed and furthermore showed comparable levels for primed cells (Fig 5G), indicating that changes in long-range chromatin interactions at the *ZBTB16* locus upon hormone induction are not maintained and likely do not explain the transcriptional memory observed.

Taken together, these results suggest that none of the chromatin changes we assayed persist after hormone washout and thus likely do not play a key role in driving transcriptional memory in *ZBTB16* priming.

### Priming increases *ZBTB16* output by increasing the fraction cells responding to hormone treatment and augmented activation by individual cells

A more robust transcriptional response by a population of cells upon a second hormone exposure can be the consequence of a larger fraction of cells responding (Fig 6A). Moreover, increased transcript levels can be achieved when individual cells respond more robustly when primed (Fig 6A). To assay *ZBTB16* mRNA expression at a single-cell level, we performed RNA FISH in A549 cells comparing primed and naïve cells. Using probes targeting the

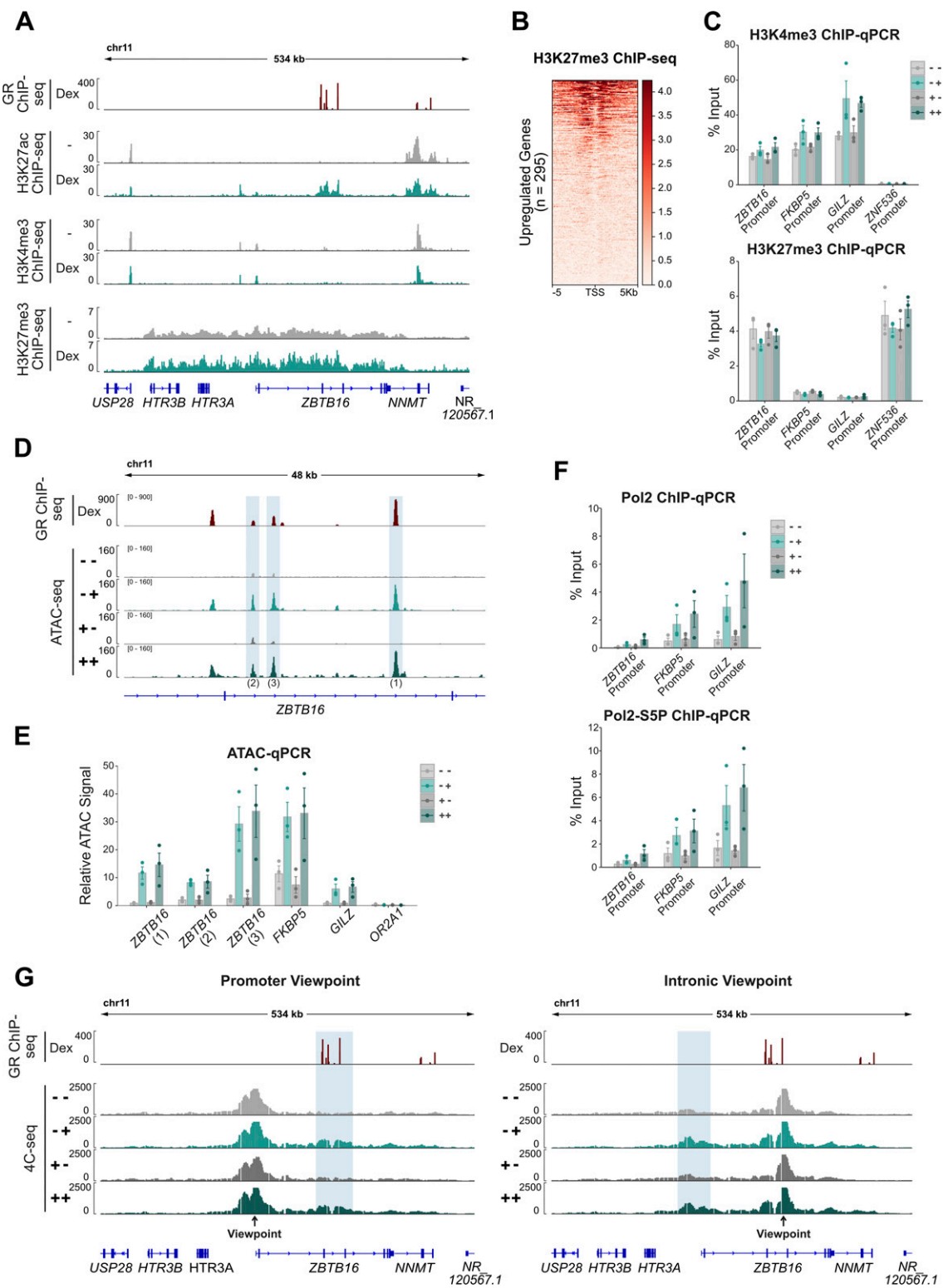

**Figure 5.   Short-term maintenance of GC-induced chromatin changes at the *ZBTB16* locus.**
**(A)** Genome browser visualization of the *ZBTB16* locus in A549 cells showing RPKM-normalized ChIP-seq signal tracks for GR (100 nM Dex, 3 h; data from reference 37),
H3K27ac (±100 nM Dex, 4 h; data from reference 41), H3K4me3 (±100 nM Dex, 4 h; data from reference 41) and H3K27me3 (100 nM Dex or EtOH, 1 h; data from reference 41).
**(B)** Heat map visualization of H3K27me3 ChIP-seq (RPKM-normalized; 1 h EtOH; data from reference 41) read coverage in A549 cells at up-regulated genes (±5 kb around
transcription start site of longest transcript variant). Heat maps are sorted by ChIP-seq signal in descending order. **(C)** ChIP-qPCR targeting H3K4me3 (top) and
H3K27me3 (bottom) at promoters of indicated genes in A549 cells. Cells were treated as described in Fig 3A. Mean % input ± SEM (n = 3) is shown. **(D)** Genome browser
visualization of GR peaks at the *ZBTB16* locus in A549 cells showing RPKM-normalized GR ChIP-seq (100 nM Dex, 3 h; data from reference 37) and normalized ATAC-seq signal

coding sequences of *ZBTB16*, we could detect individual transcripts (orange spots in the cytoplasm) as well as the sites of transcription (larger foci in the nucleus, see the Materials and Methods section), which reflect the number of actively transcribing alleles (Fig 6B). Consistent with our RNA-seq and qPCR data, *ZBTB16* levels are barely detectable for untreated cells (Figs 3E, 4A, and 6B C). This changes when cells are treated with hormone, however, increased transcript levels and transcription foci are only detectable for a minor subset of cells (Fig 6B–D). Conversely, hormone washout resulted in a complete loss of both transcription foci and cells with higher transcript levels (Fig 6B–D). The response to hormone of primed cells is changed in two ways when compared to naïve cells. First, a larger fraction of cells shows hormone-induced increases in *ZBTB16* transcript levels and transcriptional foci (Fig 6C and D). Second, a comparison of the distribution of transcripts per cell shows that a subset of the primed cells expresses *ZBTB16* at levels that are higher than for naïve hormone-treated cells (Fig 6C). These effects were even more pronounced when cells were exposed to GCs a third time (Fig 6C and D). To compare our findings for the *ZBTB16* gene with a GR-target gene that does not change its transcriptional response upon priming, we analyzed the *FKBP5* gene. Compared with *ZBTB16*, *FKBP5* is expressed at higher basal levels and accordingly its transcripts are detected for most cells regardless of whether cells were hormone-treated or not (Fig 6E and F). Upon hormone treatment, both the number of transcripts per cell and the number of transcriptional foci increases. However, in contrast to *ZBTB16*, most cells respond to hormone treatment by having more transcripts per cell and more detectable transcriptional foci (Fig 6F and G). Moreover, consistent with a lack of priming, the distribution of transcriptional foci and transcripts per cells for *FKBP5* does not show a noticeable change when comparing naïve and primed cells.

Together, we find that priming changes the number of cells responding and the robustness of the response of individual cells that are specific for the *ZBTB16* gene which together explain the more robust response observed for this gene.

## Discussion

GCs are released by the adrenal cortex in response to various types of stress. For an effective response to acute stress, cells need to react fast but should also reverse their response when the stressor is no longer present. However, when stressors are encountered repeatedly, habituation and an altered, for example, blunted, response might be crucial to ascertain an organisms well-being (48 *Preprint*). Physiologically, GCs are released in a circadian and ultradian manner (13, 14). When stress is chronic, this results in a prolonged exposure to high hormone levels and is associated with

severe pathological outcomes (10, 49). Similarly, when GCs are used therapeutically, high doses and repeated long-term exposure causes severe side effects and can result in emerging GC resistance (50, 51). Motivated by previous studies that indicated that GCs induce long-term chromatin changes, we explored genome-wide changes in chromatin accessibility and assayed if these changes persist after hormone washout. In addition, we investigated if prior exposure to GCs influences the transcriptional response to the subsequent exposure of the same stimulus. In contrast to previous studies (31, 36), we did not find convincing evidence that changes in chromatin accessibility persist after hormone washout as both GC-induced increases and decreases in chromatin accessibility universally reversed to their pre-hormone exposure levels (Figs 2B and S3B). One difference to the prior study that described persisting GR-induced hypersensitivity for more than 9 d is that they studied memory in another cell type (mouse L cell fibroblast) (36). Another difference is that we studied genome-wide changes, whereas they studied a single exogenous stably integrated mouse mammary tumor virus sequence. Hence, it is possible that their results do not represent a phenomenon which is commonly observed at endogenous mammalian loci. Thus, even though we did not find convincing persistence of changes in chromatin accessibility in either one of two cell lines tested, we cannot rule-out that cell type-specific mechanisms facilitate sustained accessibility. Moreover, given that we use immortalized cell lines for our studies and examined a relatively short single exposure to hormone, the bearing of our findings for in vivo pathological exposure of tissues to GCs is unclear. A recent study reported sustained increases in chromatin accessibility in mouse mammary adenocarcinoma cells 40 min after hormone washout (31). The reversal after the 24-h washout in our study indicates that GC-induced changes are universally short-lived and do not persist beyond one cell cycle in A549 and U2OS cells. We conclude that maintained chromatin openness as a result of GR binding is not a general trait of GR activation and rather appears to represent a cell type- and locus-specific phenomenon.

Mirroring what we say in terms of chromatin accessibility, transcriptional responses also seem universally reversible with no indication of priming-related changes in the transcriptional response to a repeated exposure to GC for any gene with the exception of *ZBTB16*. Although several changes in the chromatin state occurred at the *ZBTB16* locus, none of these changes persisted after hormone washout arguing against a role in transcriptional memory at this locus (Fig 5). Similarly, the increased long-range contact frequency between the *ZBTB16* promoter region and a GR-occupied enhancer does not persist after washout (Fig 5G). Notably, our RNA FISH data showed that *ZBTB16* is only transcribed in a subset of cells; hence, it is possible that persistent epigenetic changes occurring at the *ZBTB16* locus also only occur in a small subset of cells and could thus be masked by bulk methods such as ChIP-seq or

tracks. For the ATAC-seq experiments, the cells were treated as described in Fig 3A. **(E)** Highlighted peaks in (D) and other regions as indicated were targeted in ATAC-qPCR quantification. A549 cells were treated as described in Fig 3A. Mean ATAC signal (normalized to gDNA) ± SEM (n = 3) is shown. **(F)** Same as for (C) except that ChIP-qPCR targeting RNA PolII (top) and RNA PolII-S5P (bottom) is shown. **(G)** Genome browser visualization of the region around the *ZBTB16* locus in A549 cells showing normalized 4C-seq signal tracks with the *ZBTB16* promoter as a viewpoint (left) and an intronic GR peak as a viewpoint (right). Cells were treated as described in Fig 3A. One representative replicate of two biological replicates is shown. Regions with increased contact frequencies upon GC treatment are highlighted with blue shading.

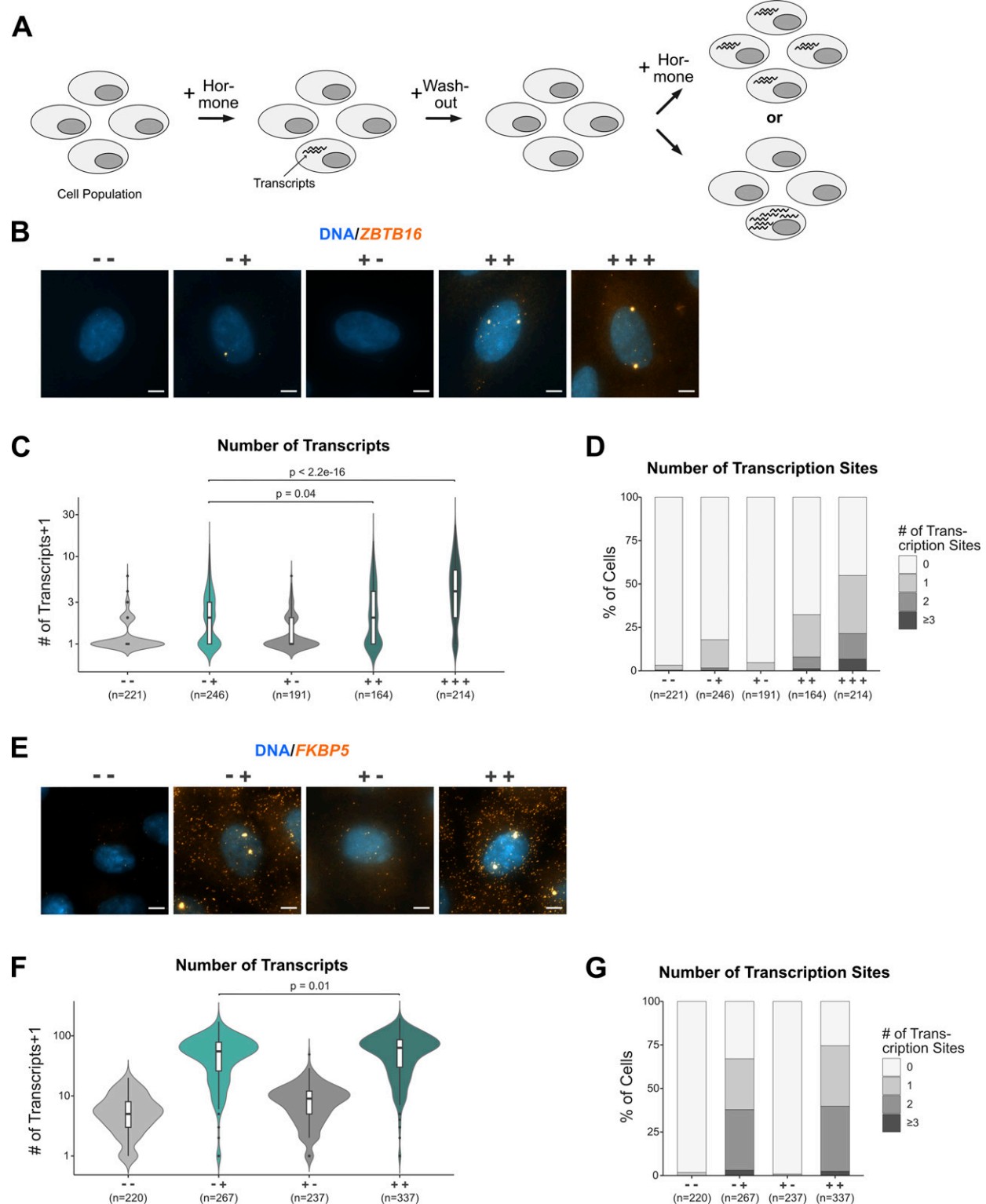

**Figure 6. Single-cell analysis comparing the activation of the ZBTB16 gene between "primed" and "naïve" cells.**
**(A)** Cartoon depicting how a more robust response can be driven by both more cells responding and by individual cells that respond more robustly. **(B)** Representative image of an RNA FISH experiment targeting *ZBTB16* mRNA (orange) in A549 cells. Nuclei were counterstained with DAPI (blue). Cells were hormone treated before fixation as described in Fig 3A. Scale bar 5 μm. **(C)** Violin plots with box plots inside showing the number of *ZBTB16* transcripts + 1 per cell as detected by RNA FISH for treatments as in (B). Results derived from three biological replicates are shown. *P*-values were calculated using a two-tailed *t* test. **(D)** Stacked bar graphs showing the percentage of cells with 0, 1, 2, or ≥3 visible transcription sites of *ZBTB16* per treatment in (B) as detected by RNA FISH. Results from three biological replicates are shown. **(E)** Same as (B), except that *FKBP5* mRNA was targeted. **(F)**. Same as (C), except that *FKBP5* mRNA was targeted. **(G)** Same as (D), except that *FKBP5* mRNA was targeted.

ATAC-seq. This could also serve as a potential explanation for the observed overlay of H3K27ac and H3K27me3 marks at the *ZBTB16* promoter (Fig 5A). Because these marks are mutually exclusive on the same histone, it is conceivable that the cells of the population that transcribe *ZBTB16* are responsible for the H3K27ac signal whereas the non-responders are decorated with H3K27me3. However, among GR-responsive genes, the presence of H3K27me3 is not a unique feature (Fig 5B) nor is the low base mean level of this gene (Fig 3C) making it unlikely that these features explain the gene-specific transcriptional memory seen for the *ZBTB16* gene. Another mechanism underlying the priming of the *ZBTB16* gene could be a persistent global decompaction of the chromatin as was shown for the *FKBP5* locus upon GR activation (33). Likewise, sustained chromosomal rearrangements, which we may not capture by 4C-seq, could occur at the *ZBTB16* locus and affect the transcriptional response to a subsequent GC exposure. Furthermore, prolonged exposure to GCs (several days) can induce stable DNA demethylation as was shown for the tyrosine aminotransferase (*Tat*) gene (52). The demethylation persisted for weeks after washout and after the priming, activation of the *Tat* gene was both faster and more robust when cells were exposed to GCs again (52). Interestingly, long-term (2 wk) exposure to GCs in trabecular meshwork cells induces demethylation of the *ZBTB16* locus raising the possibility that it may be involved in priming of the *ZBTB16* gene (53). However, it should be noted that our treatment time (4 h) is much shorter. Finally, enhanced *ZBTB16* activation upon a second hormone exposure might be the result of a changed protein composition in the cytoplasm after the first hormone treatment. In this scenario, increased levels of a cofactor produced in response to the first GC treatment would still be present at higher levels and facilitate a more robust activation of *ZBTB16* upon a subsequent hormone exposure. Although several studies have reported gene-specific cofactor requirements (54), the fact that we only observe priming for the *ZBTB16* gene would make this an extreme case where only a single gene is affected by changes in cofactor levels.

ZBTB16 is a transcription factor that belongs to POZ and Krüppel family of transcriptional repressors (55). It plays a role in various processes including limb development, stem cell self-renewal and innate immune responses (55, 56). Interestingly, both GR and ZBTB16 are linked to metabolic syndrome including changes in insulin sensitivity (57), raising the possibility that the severe metabolic side effects experienced by patients undergoing long-term GC treatment could involve mis-regulation of *ZBTB16* as a result of repeatedly prolonged GR activation. *ZBTB16* might also play a role in emerging GC-resistance given that increased levels negatively regulate GR transcriptional activity (58). Specifically, GC-induced apoptosis and gene regulation are blunted upon *ZBTB16* overexpression, whereas GC sensitivity increases when *ZBTB16* is knocked down (59).

In summary, we report that a relatively short-term GR activation results in universally reversible changes in chromatin accessibility as measured by ATAC-seq. This holds true for sites with increased chromatin accessibility but also for a large number of sites we identified that become less accessible upon GC treatment. In contrast to opening sites, closing sites are typically not GR-occupied yet are enriched near repressed genes in line with other studies (21, 22, 39 *Preprint*, 44), suggesting that transcriptional repression by GR in general does not require nearby GR binding. Given the circadian release of GCs and their role in responding to stress it makes physiological sense that changes

in chromatin accessibility and transcriptional responses are reversible under normal circumstances. However, this might be different when cells are exposed to GCs for extended periods of time, as occurs when GCs are used therapeutically. Despite the universal reversibility of GC-induced chromatin accessibility, we found a single gene that was primed by prior hormone exposure. Interestingly, this gene was only activated in a fraction of cells which is consistent with a recent single-cell RNA-seq study showing that many target genes are only regulated in subset of cells upon GC exposure (60). Potentially explaining this cell-to-cell variability, single-cell studies show that chromatin states are heterogeneous among populations of cells (61). Our single-cell studies show that priming resulted in a larger fraction of cells activating *ZBTB16* upon repeated GC treatment and also in a fraction of cells that responded more robustly. Although our bulk studies did not uncover lasting chromatin changes that could explain priming of the *ZBTB16* gene, we envision that single-cell profiling of the chromatin landscape upon repeated exposure to GCs could help further unravel mechanisms that allow individual cells to "remember" a previous hormone exposure and change their response when the signal is encountered again.

# Materials and Methods

### Cell culture and hormone treatments

A549 (CCL-185; ATCC) and U2OS-GR cells expressing stably integrated rat GRα (43) were cultured in DMEM supplemented with 5% FBS.

#### *Hormone washout treatments in A549 cells (Fig 2)*
Cells were treated with 100 nM dexamethasone (Dex), 100 nM hydrocortisone (Cort), or 0.1% EtOH (as vehicle control) for 20 h and either harvested immediately or washed 2× with PBS and subsequently cultured in hormone-free medium for 24 h before harvest.

#### *Hormone washout treatments in U2OS-GR cells (Fig S3)*
Cells were treated with 100 nM Dex or 0.1% EtOH for 4 h and then were either harvested immediately or washed 2× with PBS and subsequently cultured in hormone-free medium for 24 h before harvest.

#### *Hormone re-induction treatments (Figs 3–6)*
Cells were treated with either 100 nM Dex, 100 nM Cort, or 0.1% EtOH. After 4 h, cells were washed 2× with PBS and cultured in hormone-free medium for 24 h, after which cells were treated again with either 100 nM Dex, 100 nM Cort, or 0.1% EtOH.

"Hormone-free medium" in the above cases refers to medium containing 5% FBS with no extra hormone added. However, given that FBS contains low levels of cortisol, the medium is technically not hormone-free. Important for our study, the FBS-derived cortisol level is too low to induce discernible genomic binding or transcriptional regulation by GR.

### RNA extraction and analysis by quantitative real-time PCR (qPCR)

RNA was extracted from cells treated as indicated in the figure legends with the RNeasy Mini Kit (QIAGEN) and reverse transcribed

**Table 1. qPCR primers for the quantification of gene expression.**

| Name | Sequence 5′–3′ |
|------|----------------|
| ZBTB16 | fwd: AGAGGGAGCTGTTCAGCAAG |
| | rev: TCGTTATCAGGAAGCTCGAC |
| FKBP5 | fwd: TGAAGGGTTAGCGGAGCAC |
| | rev: CTTGGCACCTTCATCAGTAGTC |
| RPL19 | fwd: ATGTATCACAGCCTGTACCTG |
| | rev: TTCTTGGTCTCTTCCTCCTTG |
| GILZ | fwd: CCATGGACATCTTCAACAGC |
| | rev: TTGGCTCAATCTCTCCCATC |

using the PrimeScript RT Reagent Kit (Takara). qPCR was performed using primers listed in Table 1.

**Total RNA-seq**

Total RNA was extracted from 1 million A549 cells treated as indicated in the figure legends using the QIAGEN RNeasy Mini kit. Sequencing libraries were prepared with the KAPA RNA HyperPrep kit with RiboErase (#08098131702; Roche) and samples were submitted for paired-end Illumina sequencing. RNA-seq data for U2OS-GR cells (1 μM Dex or EtOH, 4 h; [62]) were downloaded from ArrayExpress (accession number E-MTAB-6738).

**ATAC-qPCR and ATAC-seq**

ATAC assays for A549 and U2OS-GR cells treated as indicated in the figure legends, were performed as previously described (63). For ATAC-seq, samples were paired-end sequenced. For ATAC-qPCR, ATAC libraries were quantified by qPCR with primers shown in Table 2.

**ChIP-qPCR and ChIP-seq**

ChIP assays for cells treated as indicated in the figure legends, were performed as previously described (64) using antibodies targeting GR (N499, 2 μl/ChIP), H3K27ac (#C15410196, 1.4 μg/ChIP; Diagenode), H3K4me3 (#C15410003, 1.4 μg/ChIP; Diagenode), H3K27me3 (#C15410195, 1.4 μg/ChIP; Diagenode), RNA Polymerase II 8WG16 (Covance #MM2-126R, 2 μg/ChIP), and Phospho-RNA pol II CTD (Ser5) (#MA1-46093, 2 μg/ChIP; Thermo Fisher Scientific). ChIP assays for qPCR quantification were performed using primers shown in Table 2. P-values were calculated using a two-tailed t test. Sequencing libraries were generated with the Kappa HyperPrep kit (#07962363001; Roche) and submitted for single-end Illumina sequencing.

For A549 cells, the following ChIP-seq datasets were downloaded from the Gene Expression Omnibus: GR (two replicates; 100 nM Dex or EtOH for 3 h; GSE79431, [37]), H3K27ac (100 nM Dex, 0/4 h; GSM2421694/GSM2421873; [41]), H3K4me3 (100 nM Dex, 0/4 h; GSM2421504/GSM2421914; [41]), H3K27me3 (100 nM Dex or EtOH, 1 h; GSM1003455/GSM1003577; [41]), and p300 (100 nM Dex, 0/4 h; GSM2421805/GSM2421479; [41]).

For U2OS-GR cells, the following ChIP-seq datasets were downloaded from the Sequence Read Archive or ArrayExpress: GR replicate

**Table 2. qPCR primers for the quantification of ATAC and ChIP experiments.**

| Name | Sequence 5′–3′ |
|------|----------------|
| ZBTB16 (2) | fwd: AACTTTGTGTGATCCCTATC |
| | rev: GGCAGTATCTAGATGGTAGC |
| DUSP1 | fwd: TACAAACAGATCTCCATGC |
| | rev: CAAATGAGGAGGTTAGACAG |
| FKBP5 | fwd: CATCACTTAAACTGGAGCTC |
| | rev: GGGTGTTCTGTGCTCTTC |
| SLC9A8 | fwd: TTCAGGAAGAATACTCAAGC |
| | rev: ACTCCCTATTGTTTCACATG |
| PTK2B | fwd: TGGGACCTAATGATTAACTG |
| | rev: AACCTAATACCCACACAGTC |
| OR2A1 | fwd: TGCATGACGCAGACCTTTCT |
| | rev: ATGAGAACCACATGGGCCAG |
| ZBTB16 (1) | fwd: ATATCCTGGACCTATCAATG |
| | rev: ACAGATTCAGGGAAGAGG |
| FGF5 | fwd: GTAGATAGCATGTACAGAGCGC |
| | rev: AATCCCATGCCTTCCTGCTC |
| CYP24A1 | fwd: TGAACCCAATTGCTCCCGTC |
| | rev: TGCCTACCCTGACAGTCATG |
| NTSR1 | fwd: AGCTCCACTTCTGATCTGTCAC |
| | rev: GTTCGATCCGGTTTGCTGAG |
| GILZ | fwd: GAGAGATTAATGCCTTTCTG |
| | rev: CCATATACTTCCGATCATTC |
| FKBP5 (2) | fwd: CTGGCCTACTTGTACACAC |
| | rev: TGCAGTAACACAATGTACAG |
| SRPK2 | fwd: GACATCACACCTCGTCTC |
| | rev: GGATGTGCTCTTCATGTC |
| ZBTB16 (3) | fwd: GCCTGTGTTTGTTATTGTAG |
| | rev: GTGTATGATGACAAACTTGG |
| ZBTB16 (4) | fwd: CTCTCCCTACTCTGAATTTG |
| | rev: ATAAACTCTCTGGAATGCTG |
| ZBTB16 promoter | fwd: GTGGGTGCTCTTATGTATG |
| | rev: ATCTACTCGTCAGCTCCTC |
| FKBP5 promoter | fwd: TACTGAACGGCGGCCAAACG |
| | rev: ATCGGGTTCTGCAGTGGTGG |
| GILZ promoter | fwd: CTCTAATCAGACTCCACCTC |
| | rev: TAGACAACAAGATCGAACAG |
| ZNF536 promoter | fwd: ATAGGATCTGGACTCAAGTG |
| | rev: AGCTGAATTACCTTGAGAAC |

1 (1 μM Dex, 1.5 h; Sequence Read Archive accession SRX256867/SRX256891; [65]), GR replicate 2 (1 μM Dex, 1.5 h; ArrayExpress accession E-MTAB-9616; [66, 67 Preprint]), H3K27ac (1 μM Dex or EtOH, 1.5 h; ArrayExpress accession E-MTAB-9617; [66]).

## 4C-seq

4C template preparation from 5 million A549 cells treated as indicated in the figure legend was done as described in reference 47 using Csp6I (Thermo Fisher Scientific) and DpnII (Thermo Fisher Scientific) as primary and secondary restriction enzymes, respectively. Sequencing library preparation was essentially performed as described in reference 68, with the exception that a 1.5× AMPure XP purification was carried out after the first PCR. Primer pairs used for the inverse PCR are listed in Table 3. 4C libraries were submitted for single-end Illumina sequencing.

## RNA FISH

The following FISH probes, labelled with Quasar 570 Dye, were purchased from Stellaris (Biosearch Technologies, Inc.): (1) human *FKBP5* (#VSMF-2130-5), and (2) human *ZBTB16* which were designed targeting the complete coding sequence of human *ZBTB16* (GenBank: BC029812.1) using the Stellaris RNA FISH Probe Designer (Biosearch Technologies, Inc.) available online at www.biosearchtech.com/stellarisdesigner (version 4.2). The RNA FISH procedure was performed on A549 cells (treated as indicated in the figure legends) according to the manufacturer's recommendations for adherent cells (www.biosearchtech.com/stellarisprotocols). Images were captured on an Axio Observer.Z1/7 (Zeiss) using a 100× Oil Immersion Objective (NA = 1.4) running under ZEN 2.3.

## Computational analysis

### *ATAC-seq*

**Data processing** Bowtie2 v2.1.0 (69) (–very-sensitive) was used to map the paired-end ATAC-seq reads to the reference human genome hg19. Reads of mapping quality <10 and duplicate reads were filtered out with SAMtools v1.10 (70) and Picard tools (MarkDuplicate) v.2.17.0 (http://broadinstitute.github.io/picard/), respectively. Reads were shifted to account for Tn5 adaptor insertion (described in reference 71) with alignmentSieve from deepTools v3.4.1 (72).

**Calling opening/closing/non-changing sites** Opening/closing/non-changing sites were determined based on the 20 h Dex-, Cort and EtOH-treated (no washout) ATAC-seq data for A549 cells and the 4 h Dex- and EtOH-treated (no washout) ATAC-seq data for U2OS-GR cells. For opening regions, peaks were called on hormone-treated samples over vehicle-treated samples with MACS2 v2.1.2 (73) (–broad –broadcutoff 0.001). For closing regions, peaks were called in vehicle-treated samples over hormone-treated samples using the same MACS2 settings. For A549 cells, called peaks were filtered for a "fold_enrichment" score of >2 and a high-confidence set of increasing/decreasing

peaks was obtained by extracting the intersect between Dex- and Cort-treated samples with BEDtools v2.27.1 intersect (74). For U2OS-GR cells, called peaks were filtered for a "fold_enrichment" score of >3. Opening and closing sites were excluded if they overlapped a promoter region (±500 bp around TSS) of any transcript variant of up-regulated or down-regulated genes, respectively.

To define sites of non-changing accessibility, peaks were called independently for each treatment using MACS2 (–broad–broadcutoff 0.001, "no control"). Next, called peaks were filtered for a "fold_enrichment" score of >8 and the intersect between all treatments was extracted and sites of increasing and decreasing accessibility were removed with BEDtools intersect. ENCODE blacklisted regions for hg19 (41) and regions within unplaced contigs and mitochondrial genes were removed from all peaks.

To define opening/closing sites before and after washout, regions of increased or decreased accessibility were called using the "24 h after washout" ATAC-seq datasets in the same way as described above for the "before washout" samples. Subsequently, overlapping sites between the "before" and "after washout" were extracted with BEDtools v2.27.1 intersect.

**Normalization for heat map and genome browser visualizations, scatter plot correlations** To account for differences in signal-to-noise ratios, ATAC-seq samples were normalized with individual scaling factors. For this purpose, peaks were called for each treatment using MACS2 v2.1.2 (–broad –broadcutoff 0.01, "no control"). To obtain a list of ATAC-seq peaks of high-confidence, the intersect of all treatments was extracted with BEDtools intersect. Having removed regions within unplaced contigs and mitochondrial DNA as well as ENCODE blacklisted regions for hg19, fragments of each sample were counted on the high-confidence regions using featureCounts (allowMultiOverlap=TRUE, isPairedEnd=TRUE) (75). The estimateSizeFactorsForMatrix function from DEseq2 v1.26.0 (76) was applied to calculate the scaling factors. The reciprocals of the resulting factors were taken and provided as scaling factors to the deepTools v3.4.1 (72) function bamCoverage to generate scaled bigWigs. Heatmaps and mean signal plots (±2 kb around the peak center) were generated with the deepTools functions computeMatrix (reference-point) and plotHeatmap, using the scaled bigWig files as input. All heatmaps are sorted by GR ChIP-seq signal in descending order. Scatter plot correlation analysis was performed with multiBigwigSummary (bins -bs 5000) and plotCorrelation (-c pearson -p scatterplot –removeOutliers) from deepTools using the normalized bigWig files as input.

## Motif analysis at opening, closing, and non-changing sites

Motif enrichment analysis at opening, closing and non-changing sites was performed with AME (77) of the MEME suite v5.1.1 (78). For

**Table 3. 4C primer sequences for inverse PCR.**

| Name | Sequence 5′-3′ |
|---|---|
| Promoter viewpoint | fwd: TACACGACGCTCTTCCGATCTAGCAGAGAGGAGTTGAGG |
| | rev: ACTGGAGTTCAGACGTGTGCTCTTCCGATCTTAGTGAACCAGGTGCCAG |
| Intronic viewpoint | fwd: TACACGACGCTCTTCCGATCTATGTGTGCGTTCATGTATGT |
| | rev: ACTGGAGTTCAGACGTGTGCTCTTCCGATCTGAGGAAAGGTTAGGAAGTGG |

A549 cells, all opening sites and an equal number of randomly sampled closing and non-changing sites were used as input. For U2OS-GR cells, all closing sites as well as an equal number of randomly sampled opening and non-changing sites were used as input. The sequences (±250 bp around the center) were scanned for the JASPAR 2018 CORE Vertebrates Clustering motifs (79) using shuffled input sequences as control. Motif hits were included in the final output if the E-value was below $10^{-30}$ for either opening, closing, or non-changing sites.

### ChIP-seq

Bowtie2 v2.1.0 (69) (–very-sensitive) was used to map ChIP-seq reads to the reference genome hg19. The GR ChIP-seq reads for rep1 (SRP020242, [65]) were mapped by setting options in Bowtie2 as "--very-sensitive -X 600 –trim5 5." SAMtools v1.10 (70) was used to remove reads of mapping quality <10. Duplicate reads were filtered out using the MarkDuplicate function from Picard tools v.2.17.0 (http://broadinstitute.github.io/picard/). To generate normalized bigWig files, in which the number of reads per bin is normalized by applying the Reads Per Kilobase per Million normalization, bamCoverage from deepTools v3.4.1 (72) was used (–normalizeUsing RPKM). Heat maps and mean signal plots (±2 kb around the peak center) were generated with the functions computeMatrix (reference-point) and plotHeatmap from deepTools. All heat maps are sorted by GR ChIP-seq signal in descending order. GR ChIP-seq peaks for each replicate were called over input with MACS2 v2.1.2 (73) setting a qvalue cutoff of 0.01. BEDtools intersect v2.27.1 (-u) (74) was used to extract overlapping peaks that were called in both replicates to obtain a final GR peak set. ENCODE blacklisted regions for hg19 (41) and regions within unplaced contigs and mitochondrial genes were removed. Overlap between opening/closing/non-changing sites and GR peaks was determined using BEDtools intersect (-u).

### RNA-seq

Paired-end reads were aligned to the hg19 reference genome using STAR v2.7.0a (80). Reads of mapping quality <10 were removed with SAMtools v1.10 (70). For genome browser visualization, triplicates were merged using the merge function from SAMtools and RPKM-normalized bigWig files were generated using bamCoverage from deepTools v3.4.1 (72).

Differential gene expression analysis for A549 cells was performed based on the quantification of read coverage in introns of the total RNA-seq data. For this purpose, the annotation file of NCBI RefSeq genes available from the UCSC Genome Browser (81) was downloaded and information on the longest transcript variants per gene was extracted. Introns of the longest transcripts were obtained with the intronicParts function from GenomicFeatures (82). Next, to ensure that the intronic regions do not overlap any mRNA sequences, exonic regions of all transcripts (obtained with the exonicParts function from GenomicFeatures) were subtracted from the introns (using the GenomicFeatures function disjoin on the combined intronic and exonic regions followed by subsetByOverlaps). Finally, introns which associated with more than one gene were excluded to ensure only unique intron regions were contained in the final set. Reads within intronic regions were counted with featureCounts (75) (isPairedEnd=TRUE, primaryOnly=TRUE, requireBothEndsMapped=TRUE, countChimericFragments=FALSE, useMetaFeatures=FALSE). Next, the intronic read counts per gene were summed. Differential expression analysis was performed using DESeq2 v1.26.0 (76).

For U2OS-GR cells, mRNA-seq data were from ArrayExpress accession number E-MTAB-6738 and differential expression analysis was carried out based on exonic read coverage. Exonic regions of the longest transcripts were obtained with the exonicParts(linked.to.single.gene.only=TRUE) and disjoin functions from GenomicFeatures (82). Read counting and differential expression analysis were performed as described above for the A549 cells.

### Linking peaks to gene regulation

Genes were grouped as up-regulated (log2 fold change > 1, adjusted P-value < 0.05, baseMean > 40), down-regulated (log2 fold change < –1, adjusted P-value < 0.05, baseMean > 40) or nonregulated (0.1 > log2 fold change > –0.1, baseMean > 40). For nonregulated genes, 500 genes were randomly sampled for A549 cells and 1,000 genes for U2OS-GR cells.

For each gene, it was determined whether at least one peak fell within a ±50 kb window around the TSS of the longest transcript variant using BEDtools intersect v2.27.1 (-u) (75). If a peak overlapped the ±50 kb window around the TSS of multiple genes, it was assigned to the closest gene. P-values were calculated performing a Fisher's exact test.

### 4C-seq

4C-seq data were analyzed with the pipe4C pipeline (68) using default settings except setting the –wig parameter to obtain WIG output files.

### RNA FISH image analysis

Raw images were processed by applying a Maximum Intensity Projection in ZEN 3.0 (Zeiss) including the full Z-Stack of 26 slices. To count transcription sites and individual transcripts, image analysis was performed in ZEN 3.0. Specifically, nuclei were detected by the DAPI staining using fixed intensity thresholds after a faint smoothing (Gauss: 2.0) and segmentation (watershed: 10), and subsequently filtering by circularity (0.5–1), size (75–450 $\mu m^2$) and a mean intensity of maximum (4,200). To define a region for the cytoplasm, a ring (width 30 pix = 3.96 $\mu m$) was automatically drawn around the nuclei. Transcripts were identified within the nuclei and surrounding rings after Rolling Ball Background Subtraction with a radius 5 pix by a fixed fluorescence threshold and subsequently filtered by area (0–0.33 $\mu m^2$). Transcription sites were identified with the same parameters as the transcripts as well as additionally filtering for an area larger than 0.38 $\mu m^2$.

## Data Availability

The ATAC-seq, RNA-seq, ChIP-seq, and 4C-seq data generated for this study were submitted to the ArrayExpress repository under the following accession numbers: E-MTAB-9909, E-MTAB-9910, E-MTAB-9911, E-MTAB-9912, E-MTAB-9914, E-MTAB-9915, E-MTAB-9923.

## Supplementary Information

## Acknowledgements

We thank Edda Einfeldt and Beatrix Fauler for excellent technical support. This work was supported by the Max Planck Society.

### Author Contributions

M Bothe: conceptualization, data curation, formal analysis, investigation, visualization, and writing—original draft, review, and editing.
R Buschow: formal analysis, methodology, and writing—original draft.
SH Meijsing: conceptualization, formal analysis, supervision, investigation, and writing—original draft, review, and editing.

### Conflict of Interest Statement

The authors declare that they have no conflict of interest.

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
