## [Reviewer comments · Life Science Alliance]

Life Science Alliance

Glucocorticoid signaling induces transcriptional memory and universally reversible chromatin changes

Melissa Bothe, Rene Buschow, and Sebastiaan Meijsing

DOI: <https://doi.org/10.26508/lsa.202101080>

Corresponding author(s): Sebastiaan Meijsing, Max Planck Institute for Molecular Genetics, Berlin, Germany

Review Timeline:

Submission Date:	2021-03-29
Editorial Decision:	2021-04-05
Revision Received:	2021-07-21
Editorial Decision:	2021-08-09
Revision Received:	2021-08-12
Accepted:	2021-08-12

Transaction Report:

April 5, 2021

Re: Life Science Alliance manuscript #LSA-2021-01080-T

Dr. Sebastiaan H. Meijsing
Max Planck Institute for Molecular Genetics, Berlin, Germany
Computational Molecular Biology
Innestrasse 63-73
Berlin 14195
Germany

Dear Dr. Meijsing,

Thank you for submitting your manuscript entitled "Glucocorticoid receptor activation induces gene-specific transcriptional memory and universally reversible changes in chromatin accessibility" to Life Science Alliance.

For a brief overview, the manuscript was reviewed at Review Commons (RC) and the authors chose to submit their paper along with the reviewers' comments and their rebuttal / revision plan in response to the reviewers' comments to Life Science Alliance (LSA). At LSA, we agree with the revision plan proposed by the authors and encourage them to submit a revised manuscript in line with what they have proposed in the pbp rebuttal. Please note that the revised manuscript will be sent back to the referees (preferably the same ones as RC) and their strong support will be required for acceptance in LSA.

Thank you for this interesting contribution to Life Science Alliance. We are looking forward to receiving your revised manuscript.

Sincerely,

Shachi Bhatt, Ph.D.
Executive Editor
Life Science Alliance
<http://www.lsjournal.org>
Tweet @SciBhatt @LSAJournal

- A letter addressing the reviewers' comments point by point.
- An editable version of the final text (.DOC or .DOCX) is needed for copyediting (no PDFs).
- High-resolution figure, supplementary figure and video files uploaded as individual files: See our detailed guidelines for preparing your production-ready images, <https://www.life-science-alliance.org/authors>
- Summary blurb (enter in submission system): A short text summarizing in a single sentence the study (max. 200 characters including spaces). This text is used in conjunction with the titles of papers, hence should be informative and complementary to the title and running title. It should describe the context and significance of the findings for a general readership; it should be written in the present tense and refer to the work in the third person. Author names should not be mentioned.

B. MANUSCRIPT ORGANIZATION AND FORMATTING:

Dear editor and dear reviewers of our manuscript,

We thank Review Commons and its three reviewers for their supportive and insightful responses to our manuscript and Life Science Alliance for considering our manuscript. Below, we provide detailed responses to the reviewers' individual comments and how we have addressed them during the revision. In short, we have made several textual changes, added references, resorted the heatmaps for a consistent sort-order across experiments, added statistical tests for key findings, performed sequence motif analysis for closing sites and further investigated the role of the H3K27me3 mark in priming of the *ZBTB16* gene.

Below please find attached our detailed responses to the reviewers' comments.

Sincerely,

Sebastiaan H. Meijnsing

Reviewer #1: **Major comments:**

The manuscript is very well written. The data is clearly presented. The methods are explained in sufficient detail with a few exceptions mentioned below, and statistical analysis are adequate. There are some concerns and suggestions about the experimental design and data presentation.

– Drug treatments. It is not clear whether the cells were previously grown on charcoal-stripped serum before hormone treatments. From methods, it seems they were grown in 5% FBS and directly treated with the hormones. Also, what "hormone-free medium" mean? Is it charcoal stripped Serum or not Serum at all?

For all experiments, the cells were grown in medium containing 5% FBS. Throughout the manuscript, "hormone-free" refers to medium containing 5% FBS without added dexamethasone. Technically, this medium is not hormone-free as FBS contains low levels of cortisol. However, the levels of cortisol from the FBS in our medium seems insufficient to elicit a transcriptional response or DNA binding by GR based on experiments comparing charcoal stripped and medium containing regular 5% FBS. However, we acknowledge that it should be made clear to the reader that growth conditions technically were not hormone-free. In the revised manuscript, we have included this information in the method section. Page 4: "... 'Hormone-free medium' in the above cases refers to medium containing 5% FBS with no extra hormone added. However, given that FBS contains low levels of cortisol, the medium is technically not hormone-free. Important for our study, the FBS-derived cortisol level is too low to induce discernable genomic binding or transcriptional regulation by GR...."

Additionally, we have explicitly stated that our naïve cells are those that have not previously been exposed to high hormone concentrations. Page 12: "... 'naïve' cells, which have not been exposed to a prior high dose of hormone, ... "

Replicates for these data sets? The ATAC and Chip-Seq should have at least 2. The concordance of the ATAC-seq and Chip-seq replicates should be described and shown in supplemental figures.

The ChIP-seq peaks for GR are the intersect of two biological replicates. This is described in the Methods section (page 8). For the ATAC data, we used two biological replicates for the vehicle treated cells (before and after washout) and treated two different hormones (dexamethasone and cortisol) as replicates. In the revised manuscript, we have added figure panels (see below, Fig S1b, S3a,d) showing the concordance between the hormone-treated samples as well as vehicle-treated samples.

ATAC-seq in A549 cells:

ATAC-seq in U2OS-GR cells:

Fig1A – The ATAC-seq HM should be clustered to show which peaks in opening/closing and unchanged peaks also have called GR chip peaks. Showing browser shots as in Fig1B is cherry picking data and can be put in a supplementary figure as an example. This is a main point of emphasis of the manuscript so show the data. The atac peaks that do overlap with GR chip peaks should be sorted by GR peak intensity. The QPCR is then only needed to confirm the quantitative changes.

This is a good suggestion which we implemented in the revised manuscript, we have added figure panels (see below, Fig S1d for A549 and Fig S2b for U2OS-GR) showing the percentage of opening/closing/nonchanging sites which overlap called GR peaks.

A549 cells:

U2OS-GR cells:

As suggested by this reviewer (and also in response to a comment by one of the other reviewers), we have re-plotted the ATAC-seq heatmaps and sorted the regions by GR ChIP-seq signal (e.g. see below, Fig 1a). Likewise, we have resorted all other heatmaps we show throughout our manuscript by GR ChIP-seq signal.

Regarding the use of browser shots, obviously these are cherry picked examples, however in our opinion they serve a purpose beyond illustrating examples of individual loci that open or close as they also give the reader an idea of the quality of the ATAC-seq data.

To show both the ATAC sites and H3K27ac sites are specific to hormone treatment, a random set of 15K peaks not in this peak set also should be shown in HMs and should not change with the treatments. Why does the H3K27ac go down in the 6768 non changing sites with dex?

As described in the previous response, we have re-sorted the H3K27ac ChIP-seq heatmap by GR ChIP-seq signal (see above, Fig 1a). The proposed group of control peaks is a basically what we included as “non-changing” peaks. Regarding reduced H3K27ac signal upon Dex treatment at non-changing sites: Notably, this comparison is based on a single ChIP-seq replicate. In our experience, ChIP-seq experiments show quite some variability between biological replicates, which limits our ability to compare signal levels quantitatively. Thus, the difference could simply reflect a difference in ChIP efficiency between the treated and untreated cells. Alternatively, it could be that there is a general redistribution of H3K27ac signal towards GR-occupied opening sites. To pin down which of these explanations is valid, we would need to perform additional experiments, e.g. using spike-ins. However, this is beyond what we can do at the moment and therefore, we have revised the text to make sure that the interpretation of these results is speculative. Page 10: “... In addition, since H3K27ac levels appear to show a modest decrease at sites of non-changing accessibility one could speculate that GR activation induces a global redistribution of H3K27ac (Fig. 1a). ...”

The D & E parts of Fig1 can then be eliminated to become parts of Fig1A. Its not clear in the text that the HMs in Fig1 are all sorted in the same way.

In the revised manuscript, we have moved the GR and H3K27ac ChIP-seq heatmaps from d&e and merged them with panel a (see above: Fig 1a). As described in the response above, throughout the manuscript all

heatmaps are now sorted by GR ChIP-seq signal (e.g. see heatmaps above). This information is also provided in the figure legends e.g. for Fig 1a: "...Heatmaps are sorted by GR ChIP-seq signal in descending order..."

- Fig. 1b (and d). The ChIP data is from 3h-hormone treatment while the ATAC-seq data is from a 20h hormone treatment. It seems a bit misleading to directly compare GR occupancy with the state of the chromatin at different time windows. Shouldn't the authors show their ATAC-seq 4h treatment data (shown in Fig S1) here instead?

We agree that the 4h time-point would be the more logical choice. However, the data for the 4h time point of A549 cells only yielded a small number of peaks with significant changes upon dex treatment (possibly due to lower data quality). Therefore, we decided to use the ATAC-seq data for the 20h treatment, which is also the treatment time for the wash-out experiments. We have added a heatmap to plot the data for the 4h treatment (Fig S1a) which shows that the three categories of sites we defined based on the data for cells treated for 20h show the same overall response to hormone treatment after 4h or treatment. Finally, the data for the U2OS-GR cell line is for cells treated for 4h (Fig. S2). The results for this cell line are consistent with those found for A549 cells further indicating that the conclusions drawn on data for 20h-treated cells are likely valid.

- Fig. 1f. The authors state "downregulated genes only show a modest enrichment of GR peaks". However, there is a significant enrichment of GR-peaks in repressive genes compared to non-regulated genes. It would be interesting to see how some of these peaks look in a browser shot. While the general conclusion "transcriptional repression, in general, does not require nearby GR binding", seems valid, the observation that many GR peaks appear directly bound to nearby repressed genes ought to be more emphatically recognized in the text.

This is a fair point and was also raised by the other reviewers. During the revision, we have made textual changes to acknowledge that GR binding is enriched near repressed genes, albeit clearly to a lesser degree than for activated genes. Page 10: "...GR peaks are also enriched near repressed genes (Fig. 1d, S1f) indicating that for some genes repression might be occupancy-driven. However, the enrichment is markedly lower than for upregulated genes and repressed genes are not enriched for genes with multiple GR peaks (Fig. S1g). Similarly, analysis of a U2OS cell line stably expressing GR (U2OS-GR, [39]) showed a modest enrichment of GR peaks near downregulated genes (Fig. S2d) and no clear enrichment of repressed genes with multiple GR peaks (Fig. S2e).....".

In addition, we have added a genome browser shot of a downregulated gene with nearby GR peak (see below, Fig. S1f).

- Concept of naïve cells (Fig. 3A). If cells are normally grown in serum-containing media, which is known to have some level of steroids, can the cells described here as "Basal expression" be truly free of a primed state? In the first part of the experimental design (+/- 4h hormone), which type of media is present here? Is it 5% FBS? A concern is that the authors may require the assumption that the (4h + 24h) period is sufficient to erase all memory of the cells, which is exactly what they are trying to test.

See our response to the first major comment above.

It would be interesting to do a time course of the hormone-free period of the washout to determine the memory of the chromatin environment that results in the enhanced transcriptional response instead of just 24 and 48 hrs in A549 cells.

We agree that that would be interesting but this is something that we cannot include for now.

Fig 5A appears to show H3K27ac overlaying H3K27me marks near the promoter of ZBTB16 and at

the GR sites within the gene locus with no reduction in H3K27me levels. This seems counterintuitive and should be explained or addressed especially since the authors use quantitative comparisons of H3K27ac levels with and without treatment in other figures.

A trivial explanation for the overlaying H3K27ac and H3K27me3 marks at the *ZBTB16* locus is that the ChIP results represent a population average. From our single-cell FISH experiments, we found that only a subset of cells activates *ZBTB16* expression upon hormone treatment. Thus, a potential explanation is that the cells of the population that respond are responsible for the H3K27ac signal whereas the non-responders are decorated with H3K27me3. We have included this information in a revised discussion. Page 15:

"... This could also serve as a potential explanation for the observed overlay of H3K27ac and H3K27me3 marks at the ZBTB16 promoter (Fig. 5a). Since these marks are mutually exclusive on the same histone, it is conceivable that that the cells of the population that transcribe ZBTB16 are responsible for the H3K27ac signal whereas the non-responders are decorated with H3K27me3...."

Showing the changes of ZBTB16 upon 2nd stimulation via FISH is not terribly surprising and is even the most expected reason for higher RNA levels. Why does it only occur at that gene is a better question and is touched on in the discussion. It is more likely that this gene has a very low level of pre-hormone transcription compared to FKBP5 (see Fig 3e and the FISH images). ZBTB16 is in the lower 3rd of basemean RNA levels of GR responsive genes according to the RNAseq data. Selection of 1 or 2 other genes with similar basemean levels of RNA (from the RNA-Seq data) would make the data more....

When compared to *FKBP5*, *ZBTB16* indeed has very low levels of pre-hormone expression. However, this is unlikely to explain the observed "memory" for *ZBTB16* given that there are other genes with similarly low pre-hormone levels that do not show more robust responses upon repeated hormone exposure (see Fig. 3B,D). For the FISH experiments, we decided to include a non-primed gene (*FKBP5* as control). We agree that adding additional control genes with comparable basemean levels would be informative. For example, this would tell us if a response of only a subset of cells in the population to hormone is specific to *ZBTB16*. Based on single cell studies by others (PMID: 32170217), most GR target genes show a response in only a subset of cells indicating that this is unlikely a unique feature of *ZBTB16* explaining the priming observed. Rather than performing additional experiments, we have revised the discussion to acknowledge the difference in basemean and the potential role of cell-to-cell variability in explaining the observed "memory" for the *ZBTB16* gene. Page 15: *"...Notably, our RNA FISH data showed that ZBTB16 is only transcribed in a subset of cells, hence, it is possible that persistent epigenetic changes occurring at the ZBTB16 locus also only occur in a small subset of cells and could thus be masked by bulk methods such as ChIP-seq or ATAC-seq. This could also serve as a potential explanation for the observed overlay of H3K27ac and H3K27me3 marks at the ZBTB16 promoter (Fig. 5a). Since these marks are mutually exclusive on the same histone, it is conceivable that that the cells of the population that transcribe ZBTB16 are responsible for the H3K27ac signal whereas the non-responders are decorated with H3K27me3. However, among GR-responsive genes, the presence of H3K27me3 is not a unique feature (Fig. 5b) nor is the low basemean level of this gene (Fig. 3c) making it unlikely that these features explain the gene-specific transcriptional memory seen for the ZBTB16 gene...."*

****Minor comments:****

- In the Intro (paragraph two), the authors explain the different mechanisms by which GR might repress genes. One alternative the authors appear to have missed is the possibility of direct binding to GREs while, for example, recruiting a selective corepressor such as GRIP1 (Syed et al., 2020). There are many recent critics to the notion that transrepression via tethering is responsible for GR repressive actions at all (Escoter-Torres et al., 2020; Hudson et al., 2018; Weikum et al., 2017).

We are aware of these studies and agree that they should be included when listing the possible mechanisms by which GR can repress genes. We added the following sentence to address this issue (page 3): *"...Further, direct DNA binding by GR, for example to AP-1 response elements, and the recruitment of corepressors like GRIP is linked to the transcriptional repression of associated genes (PMID 28591827, PMID 32619221, PMID 22753499)....."*

- When the authors introduce the concept of tethering to AP-1, they go way back to the first description of tethering. However, one of the references (Ref 20) actually goes against the tethering model as they did not detect protein-protein interactions between AP-1 and GR, and also, they conclude that repression requires the DNA-binding domain.

Tethering and a role of the DNA binding domain are not mutually exclusive as the DNA binding domain might serve as the interaction interface of GR with the protein that tethers it to the DNA. We decided to

reference some of the original papers that established tethering as a possible mechanism that mediates repression rather than a review or more recent papers on the topic.

-Figure 2. The authors state "This suggests that the few sites with persistent opening are likely a simple consequence of an incomplete hormone washout and associated residual GR binding". The authors should check the subcellular distribution of GR after their washout protocol. If the washout is not completed, GR should still be in the nuclear compartment.

The careful phrasing here was to include the possibility that GR might bind DNA even when hormone is completely washed out. However, a more likely explanation is that the washout is incomplete. The residual genomic GR binding we find in our ChIP assays (Fig. S3f) shows us that a subset of GR is indeed still chromatin-bound which implies that some GR is still in the nuclear compartment.

- The first part of the manuscript (Repression through "sqelching") seems a bit disconnected from the rest of the results (reversibility in accessibility). The abstract is structured in a way that this disconnection seems much less obvious. Perhaps the authors could try to present their sqelching part in the middle of the manuscript, following the flow of the abstract? This is just a suggestion.

While revising the manuscript, we did not find a straight-forward way to implement this suggestion.

- Figures have CAPS panel letters (A,B,C, etc) while the text calls for lower case letter (a,b,c...)

In the revised manuscript, we have fixed this issue and changed all letters to lower case in the figures and text.

Reviewer #2: ****Major Comments****

1. The authors used the cancer cell lines A549 and U2OS-GR as model systems the latter additionally overexpresses GR. In order to make the work more translatable an in-vivo model comparing the effect of long-term, short-term and repeated glucocorticoid (GC) treatment on DNA accessibility and gene expression is necessary. The authors should clearly emphasize this limitation of their study in the discussion or add in-vivo data (e.g. qPCRs) to strengthen the translatability.

We agree that long-term and repeated GC treatment would be very interesting to study and would yield insights that are more likely to be relevant to, for example, emerging GC-resistance during therapeutic use especially when conducted *in vivo*. We are certainly aware of the limitations of our study and have added to the following statement to the discussion to point out the speculative nature of translating our findings to an *in-vivo* setting (Page 15): "...Moreover, given that we use immortalized cell lines for our studies and examined a relatively short single exposure to hormone, the bearing of our findings for *in vivo* pathological exposure of tissues to GCs is unclear.....".

2. The authors draw conclusions of the association of DNA accessibility, H3K27ac, P300 and GR occupancy from independent heatmaps. This cannot be easily done from the current way the data is presented. A direct link between accessibility, H3K27ac and mRNA expression of the associated gene for example is missing. a.) The authors show several heatmaps to indicate changes in accessibility, H3K27ac and P300 upon Dex treatment as well as GR binding patterns in Fig. 1 and S1. Those are sorted by decreasing signal strength (I assume). To make those results more comparable, I suggest to sort them all in the same way (e.g. by descending ATAC-Seq signal or fold-change).

As suggested by this reviewer (and also in response to a comment by one of the other reviewers), we have re-plotted the ATAC-seq heatmaps and sorted the regions by GR ChIP-seq signal (e.g. see below, Fig 1a). Likewise, we have resorted all other heatmaps by GR ChIP-seq signal.

Moreover, have added the following figure panels (Fig S1d for A549 and Fig S2b for U2OS-GR) that show that about 50% of the opening sites are GR-occupied whereas this number is markedly lower for closing sites.

A549 cells:

U2OS-GR cells:

b.) In line with a.), it is unclear to the reader if those sides opening /closing are the same sides showing increased/decreased H3K27ac or P300 occupancy and if those sides bind GR. Integrating this data together with mRNA e.g as correlation plots would strengthen the author's argument that accessibility, H3K27ac and mRNA changes are indeed correlated. What about the GR binding sites that do not change accessibility or H3K27ac? What makes those different?

Therefore, the statement "Furthermore, closing peaks, which show GC-induced loss of H3K27ac levels and lack GR occupancy (Fig. S1c-f), were enriched near repressed genes" on page 10 as well as the statement "suggesting that transcriptional repression by GR does not require nearby GR binding." in the abstract and discussion cannot be made from how the data is presented.

By resorting the data by GR ChIP-seq signal as suggested, it is now easier to connect the different types of data presented. For example, for opening sites with high levels of GR occupancy it can be seen that H3K27ac levels increase upon Dex treatment (Fig. 1a) which coincides with increased P300 levels (Fig. S1e). GR binding sites that do not change accessibility or H3K27ac already are highly accessible, have high levels of H3K27ac and have well-positioned nucleosomes flanking the GR-occupied site prior to hormone exposure (Fig. 1a, S2a). This indicates that these enhancers are already highly active before GR activation and suggests that the added binding of GR does not "boost" activity levels even further possibly due to redundancy between the actions of TFs already present and GR. To make sure the statements are substantiated by the data, we have refined the first statement to the following: Page 10-11: "..... Furthermore, closing peaks, which show GC-induced loss of H3K27ac levels, mostly lack GR occupancy and enrichment of the consensus recognition motif (Fig. S2a-c), were enriched near repressed genes.....". As a group, closing sites are enriched near down-regulated genes (Fig. 1d, S2d) and mostly lack nearby GR binding. Further, we have added a panel to figure 1 (Fig. 1b) showing the repressed gene *FGF5* which harbors a closing site lacking GR. Based on this, we think the carefully phrased sentence in the abstract is consistent with the findings presented: "suggesting that transcriptional repression by GR does not require nearby GR binding".

c.) Several recent studies have shown that GR's effect on gene expression and chromatin modification at enhancers might be locus-/context-specific ("tethering", competition, composite DNA binding) and/or recruitment of different co-regulators (see Sacta et al. 2018 (doi: 10.7554/eLife.34864), Gupte et al. 2013 (doi.org/10.1073/pnas.1309898110) and many more). Defining the GR-bound or opening/closing sides in terms of changing H3K27ac (or having H3K27ac or not) more closely would help to link those to gene expression changes e.g. in violin plots. Furthermore, the authors could include a motif analysis to see if the different enhancer behaviours can be explained by differences in the GR motif sequence or co-occurring motifs. Thereby more closely defining the mechanism of chromatin closure a sites that lack GR binding e.g. by displacement of other transcription factors as described for p65 in macrophages (Oh et al. 2017 (doi.org/10.1016/j.immuni.2017.07.012)).

In general a more detailed analysis of the data is required before the authors could state "Instead, our data support a 'squelching model' whereby repression is driven by a redistribution of cofactors away from enhancers near repressed genes that become less accessible upon GC treatment yet lack GR occupancy." on page 10. The results might also be explained by competitive transcription factor binding, tethering or selective co-regulator recruitment (e.g. HDACs).

We have investigated the link between GR occupancy, changes in H3K27ac and gene regulation in a recent study (PMID 29385519). In short, we found that GR-occupied regions near upregulated genes show bigger increases in H3K27ac than their GR occupied regions near genes that do not change expression upon hormone treatment.

In the revised manuscript, we have included a motif enrichment analysis requested in which we scanned sites which open, close or remain unchanged for the presence of JASPAR clustered motifs (see below, Fig S1c, Fig S2c). Notably, the GR consensus motif (cluster 15) was not enriched at closing sites which is in line with our observations that GR mostly does not bind to these sites.

A549 Cells:

U2OS-GR cells:

We agree that the “squenching model” is just one of several explanations for repression. We have revised the text to acknowledge that for now this is just one of several possible explanations for what we observe. Page 11: “...Taken together, our results further support a model put forward by others [23,24,61,64] in which transcriptional activation by GR is driven by local occupancy whereas transcriptional repression, in general, does not require nearby GR binding. Instead, our data are consistent a ‘squenching model’ whereby repression is driven by a redistribution of cofactors away from enhancers near repressed genes that become less accessible upon GC treatment yet lack GR occupancy. However, repression may also be driven by other mechanisms....”.

3. The authors use U2OS-GRa cells as a second cell line. Those cells overexpress rat GRa (see DOI: 10.1128/mcb.17.6.3181) in a cell line that usually does not express GR. I am wondering to what extent the overexpression reflects residence times and GR binding kinetics of cells endogenously expressing GR (mostly to at a lower protein level). At least the number of GR binding sites as well as the number of opening chromatin sites is much higher in U2OS-GR cells than A549 cells. The authors should discuss this point with respect to the observed preservation of some GR-binding sites U2OS-GR cells after Dex treatment and washout.

We have added this information to the section where we discuss possible explanations for the persistent opening of some sites in U2OS cells. Page 11–12: “...In contrast, GR still occupied persistent opening sites 24 hours after hormone withdrawal (Fig. S3e,f) indicating that the sustained accessibility is likely the result of residual GR binding to those sites. This could be linked to the fact that the U2OS cells we used in our study overexpress GR and accordingly have a higher number of GR binding sites and opening sites than A549 cells. Another explanation for the sustained occupancy at persistent loci is that a small fraction of GR is still hormone-occupied despite the 24-hour washout and dissociation half-life of Dex of approximate 10 minutes [65]. Interestingly, at low (sub- K_d), Dex concentrations, GR selectively occupies a small subset of the genomic loci that are bound when hormone is present at saturating concentrations [66]. To test if GR preferably binds at persistent sites at low hormone concentrations, we performed GR ChIPs at different Dex doses (Fig. S3g) and found residual occupancy at persistent opening sites at very low Dex concentrations (0.1 nM), whereas GR occupancy was no longer observed at sites with transient opening (Fig. S3g). This suggests that the few sites with persistent opening are likely a simple consequence of an incomplete hormone washout and associated residual GR binding.....”.

4. In figure 1 and S1, the authors show coverage plots on top of the heatmaps to show the mean signal in ATAC-Seq, GR, H3K27ac or GR signal between the different subset. These plots are statistically inappropriate as a significant portion of the enhancers does not have a signal and a few enhancers show a very strong signal (at least for H3K27ac, P300 and GR) which skews the mean. Plotting the signal distribution or the distribution of the Dex-dependent change in signal (fold-change, e.g. as violin plots) more accurately reflects the diversity in the signal response.

We agree that the coverage plots do not take the fraction of binding sites with signal into account. However, by also showing the heat maps, this information is also available to the reader. In our opinion, the coverage plots provide a straight-forward way to compare the signal for the different categories of peaks. Even though violin plots are an interesting alternative way to present the data, we believe that the heatmaps shown provide sufficient information and capture the diversity within each group.

5. CHIP qPCRs against histone marks in figures 5B and S2C are not normalized for histone H3, but the author's clearly see changes in nucleosomal occupancy at those sites by ATAC-Seq. Additional normalization by total H3 is highly recommended.

We see your point. However, based on the ATAC-signal (Fig. 5D) the changes in nucleosomal occupancy upon GC treatment are the same for naive and primed cells and revert to their base-line level after hormone withdrawal. This indicates that these loci have comparable nucleosome occupancy after wash-out. Yet, the levels for these histone modifications do not differ between primed and naive cells indicating that these histone marks do not “mark” the promoter of primed genes after wash-out.

6. Figures 1C, 2D, 4A/B, 5B/C/E, 6C/F, S2C/E and S3A-D lack statistics.

We are reluctant to put p-values on every chart, especially for experiments with few replicates. Importantly, we always plot the values for each individual data point, so the reader can gage if they differ between conditions. We have added p-values to various figures *e.g.* to figure 4 which now includes the results of statistical tests that support our claim that *ZBTB16* is primed whereas other GR target genes are not.

7. In figure 6, the authors compare the *ZBTB16* locus with *FKBP5*, a locus that as by the data presented is very different from the *ZBTB16* locus in terms of expression level (Fig 6C/F) and H3K27me3 occupancy (Fig. 5B). The authors should compare *ZBTB16* to a locus with similar expression level and H3K27me3 deposition. Especially the co-occurrence of H3K27me3 and H3K4me3 (Fig. 5B) at the *ZBTB16* promoter indicates its poised chromatin state whereas the *FKBP5* promoter is marked by an active chromatin state.

A similar suggestion was brought up by reviewer #1, here is the response we gave to this comment: When compared to *FKBP5*, *ZBTB16* indeed has very low levels of pre-hormone expression. However, this is unlikely to explain the observed “memory” for *ZBTB16* given that there are other genes with similarly low pre-hormone levels that do not show more robust responses upon repeated hormone exposure (see Fig. 3B,D). For the FISH experiments, we decided to include a non-primed gene (*FKBP5* as control). We agree that adding additional control genes with comparable basemean levels would be informative. For example, this would tell us if a response of only a subset of cells in the population to hormone is specific to *ZBTB16*. Based on single cell studies by others (PMID: 32170217), most GR target genes show a response in only a subset of cells indicating that this is unlikely a unique feature of *ZBTB16* explaining the priming observed. Rather than performing additional experiments, we have revised the discussion to acknowledge the difference in basemean and the potential role of cell-to-cell variability in explaining the observed “memory” for the *ZBTB16* gene. Page 15: “...Notably, our RNA FISH data showed that *ZBTB16* is only transcribed in a subset of cells, hence, it is possible that persistent epigenetic changes occurring at the *ZBTB16* locus also only occur in a small subset of cells and could thus be masked by bulk methods such as ChIP-seq or ATAC-seq. This could also serve as a potential explanation for the observed overlay of H3K27ac and H3K27me3 marks at the *ZBTB16* promoter (Fig. 5a). Since these marks are mutually exclusive on the same histone, it is conceivable that that the cells of the population that transcribe *ZBTB16* are responsible for the H3K27ac signal whereas the non-responders are decorated with H3K27me3. However, among GR-responsive genes, the presence of H3K27me3 is not a unique feature (Fig. 5b) nor is the low basemean level of this gene (Fig. 3c) making it unlikely that these features explain the gene-specific transcriptional memory seen for the *ZBTB16* gene....”.

8. *ZBTB16* itself is a transcriptional regulator, but its elevated expression upon repeated Dex treatment does not affect other genes. How do the authors explain this observation? Is *ZBTB16* elevated on the protein level as well?

The fact that we do not observe elevated expression of other genes upon repeated expression could be due to the relatively short length of the hormone treatment, 4 hours, which was chosen to enrich for direct target genes of GR. These four hours might be insufficient for transcription, translation and ultimately gene regulation by the *ZBTB16* protein. We have not looked at *ZBTB16* protein levels.

****Minor Comments****

1. The authors nicely explained the data analysis of their ATAC-Seq data, I recommend to include some more information on if and how the ChIP-Seq data was normalized (library size, scaling factors or spike-ins) even if most of the data sets are published.

We have clarified this information in a revised version of the manuscript. Page 8:
“... To generate normalized bigWig files, in which the number of reads per bin is normalized by applying the Reads Per Kilobase per Million normalization, bamCoverage from deepTools v3.4.1 [52] was utilized (--normalizeUsing RPKM). ... ”

2. In figures 1F and S1F, the authors show the association of opening/closing at non-changing sites

and GR peaks with genes that are up/down-regulated or unchanged upon Dex treatment. This gene-centric analysis is skewed by the different sizes of up-/down regulated gene sets and opening/closing chromatin (especially for the U2OS-GR cells that have 15.6x more opening sites than closing sites). Could the authors also include a peak-centric view showing how many closing/opening and non-changing sites are associated with down/up-regulated or unchanged genes? How good is the association (correlation)?

Based on a previous study we conducted (PMID: 29385519), we expected that binding is a poor predictor of gene regulation of nearby genes, especially for repressed genes, which is essentially what we also found here when we performed the peak-centric view requested. In line with the gene centric view, the percentage of genes near opening sites that is upregulated is higher than the percentage of genes that is downregulated. Similarly, upregulation of genes near GR peaks is more than down regulation, whereas the link between closing sites and gene regulation is rather fuzzy. Here are the numbers:

A549 cells:

2934 opening peaks

- 5.5% of nearby genes upregulated
- 0.3% of nearby genes downregulated

4635 closing peaks

- 1.6% of nearby genes upregulated
- 1.7% of nearby genes downregulated

6768 nonchanging peaks

- 2% of nearby genes upregulated
- 0.6% of nearby genes downregulated

12182 GR peaks

- 5.6% of nearby genes upregulated
- 0.7% of nearby genes downregulated

U2OS cells:

32893 opening peaks

- 5.4% of nearby genes upregulated
- 1.8% of nearby genes downregulated

2592 closing peaks

- 4.4% of nearby genes upregulated
- 7.1% of nearby genes downregulated

15806 nonchanging peaks

- 3.4% of nearby genes upregulated
- 2.8% of nearby genes downregulated

39786 GR peaks

- 6.2% of nearby genes upregulated
- 1.6% of nearby genes downregul

Since the above results mostly showed the overall pattern we found in our initial gene-centric analysis (previously Fig. 1f, S1f → now Fig. 1d, S2d) and the percentages were very low, we decided to not include them in the revised manuscript.

3. In the figures 1F and S1F it is unclear how the authors handled genes with associated peaks (within +/-50kb) that show different characteristics e.g. a gene with a peak that gains and another peak that loses accessibility. How do the authors account for >1 opening or closing peaks per gene? In relation to this. Do opening/closing sites cluster around up/down-regulated genes? What is the stoichiometry as 1.6x more closing sites (then opening sites) relate to 1/3 of repressed when compared to activated genes?

In our analysis, we looked at opening and closing peaks independently. If a peak is in the vicinity of multiple genes, it will only be assigned to the closest one. Thus, genes that have both an opening and a closing peak in the 50kb window will be included in both the analysis of closing sites and opening sites. In our original submission, we simply lumped all genes with a peak in the 50kb window together, regardless of the number of peaks. In the revised manuscript, we now also plot the number of peaks within +/- 50 kb of the TSS of genes for each gene with a peak (see below, Fig. S1g, S2e). In short, we find that genes that are upregulated are more likely to contain multiple opening sites and GR peaks than genes that are repressed or not regulated. Even though more repressed genes have a closing site nearby (Fig. 1d), the number of closing peaks does not seem markedly different for the three categories of genes (see below, Fig. S1g, S2e).

A549 cells:

U2OS-GR cells:

4. The authors claim on p10 that "We could validate several examples of opening and closing sites and noticed that opening sites are often GR-occupied whereas closing sites are not occupied by GR". As most of the ChIP-Seq experiments were performed on formaldehyde-only fixed cells, the authors might miss "tethered" sites, which are mostly linked to gene repression. You might rephrase this part to most closing sites lack direct DNA binding.

Even though several studies indicate that tethered binding can be captured using formaldehyde-only fixed cells (e.g. PMID: 32619221, PMID: 15879558), we agree that the ChIP-assay might have blind spots. We have revised the results section where we discuss these findings to the following, page 10: "...For a systematic analysis of the link between GR occupancy and chromatin accessibility changes, we integrated the ATAC-seq results with available GR ChIP-seq data [43]. This analysis showed that GR binding is observed at the majority of opening sites (Fig. 1a, S1d). In contrast, only a minor subset of closing sites shows GR ChIP-seq signal (Fig. 1a, S1d). In line with these observations, motif enrichment analysis revealed that the GR consensus recognition motif was enriched at opening yet not at closing sites (Fig. S1c) indicating that most closing sites lack direct DNA binding by GR....."

5. The P300 ChIP-Seq in Fig S1B shows less sites with P300 occupancy than sites with H3K27ac. Is this a ChIP quality issue or do other factors mediated changes in H3K27ac? Similar to mayor comment 1a, are the P300 sites on the top the same sites as the top H3K27ac sites?

This might be related to comment #4: P300 is brought to the DNA by other transcription factors, whereas H3K27ac is directly DNA-bound. This likely influences the cross-linking efficiency and consequently the ChIP signal. Additionally, P300 is not the only enzyme that can deposit the H3K27ac mark. Specifically, CREBBP (a.k.a. CBP) can also acetylate H3K27 which could explain why p300 signal only overlaps with a subset of the regions marked by H3K27ac. In the revised manuscript, we have re-sorted both the p300 and the H3K27ac heatmap (Fig. 1a, S1e) by GR ChIP-seq signal to make it easier to assess the overlap between changes in p300 occupancy and changes in H3K27ac levels. In short, the plots indicate that p300 signal and H3K27ac levels correlate.

6. Please indicate the primer position of qPCR primers if the genome browser tracks are displayed. That makes the comparison of sequencing and qPCR results easier.

In the revised manuscript, we have indicated primer positions with an asterisk (e.g. see Fig 1b). Since most genome browser shots were very zoomed out, it was not possible to show the position of individual primers.

7. The authors nicely show that GR binding sites with persisting accessibility after Dex treatment and washout in U2OS-GR cells show residual GR binding and are bound by GR at Dex concentrations of 0.1nM. Could the authors specify if differences in the GR motif exist between those and the non-persisting sites?

We have not looked into this, but a previous study by the Reddy lab (PMID: 22801371) has investigated binding sites in A549 cells that are occupied at very low Dex concentrations. They found that this is not driven by a specific GR motif but rather by the presence of binding sites for other transcription factors and chromatin accessibility.

8. The authors focus on ZBTB16, FKBP5 and GILZ to show the priming effect of glucocorticoid treatment on ZBTB16 (Fig. 4), but GILZ was not included in the initial ATAC-Seq (Fig. 1) and ATAC-Seq washout (Fig. 2) experiments. For better comparison, I recommend adding qPCR results on GILZ in figures 1 and 2.

In the revised manuscript, we have included information for *GILZ* in the ATAC-qPCR figure for the washout experiments (see below, Fig. 2d).

9. The authors indicate that the washout of Dex does restore gene expression in A549 cells to pre-Dex levels (Fig. 4). These cells did not show any persisting GR binding, so. How does the gene expression in U2OS cells behave? E.g. for the genes displayed in Fig. S2C.

This is shown in figure S4c (previously S3c) and shows that expression levels of certain genes (*ZBTB16* and *FKBP5* but not *GILZ*) stay high after Dex washout (but not cortisol wash-out) consistent with persistent GR binding at a subset of GR-occupied loci for the experiments using Dex.

10. In Fig. S3C, the authors observe that *Gilz* expression in U2OS-GR cells is similarly induced upon 1st and 2nd stimulation with Dex using 4hrs treatment. How does this relate to the preserved Dex response after 20hrs treatment and washout (Fig. S2C)? Was the expression of *GILZ* altered after 20hrs (see comment 9)? Are H3K27ac and GR signal after 4hrs Dex stimulation and washout comparable as well? Please comment on the differences observed between the 20hrs and 4hrs experiments.

For both S3e and S4c (previously S2c and S3c), cells were treated for 4h with Dex before the wash-out so we are not comparing 4h and 20h here. For the *ZBTB16* and *FKBP5* genes, the persistent GR binding after wash-out is accompanied by a preserved Dex response after wash-out. For *GILZ*, GR binding at one of the peaks near the *GILZ* gene is also preserved, yet the expression of this gene reverses to its pre-treatment levels after wash-out. A possible explanation is that the residual binding at the *GILZ* gene is observed for only one of several nearby GR peaks. Previous studies, where we deleted GR binding sites near the *GILZ* gene, have shown that the combined action of multiple GR-occupied regions is needed for robust induction of this gene (PMID: 29385519).

11. The GR enhancer of *ZBTB16* seems to be simultaneously marked H3K27ac and H3K27me3 (Fig. 5A). Please comment. Is this an artefact of bulk ChIP-Seq? Is this due to the different timings (H3K27me3 after 1h and H3K27ac after 3hrs)? Can both marks co-exists or do they reflect allelic differences?

A trivial explanation for the overlaying H3K27ac and H3K27me3 marks at the *ZBTB16* locus is that the ChIP data represents a population average. From our single-cell FISH experiments, we found that only a subset of cells activates *ZBTB16* expression upon hormone treatment so a potential explanation is that the cells of the population that respond are responsible for the H3K27ac signal whereas the non-responders are decorated with H3K27me3. On a single histone, H3K27me3 and H3K27ac are mutually exclusive. However, given that a nucleosome has 2 copies of histone H3, both modifications can in principle co-exist. We have included this information in a revised discussion. Page 15:

"... This could also serve as a potential explanation for the observed overlay of H3K27ac and H3K27me3 marks at the ZBTB16 promoter (Fig. 5a). Since these marks are mutually exclusive on the same histone, it is conceivable that that the cells of the population that transcribe ZBTB16 are responsible for the H3K27ac signal whereas the non-responders are decorated with H3K27me3. ..."

12. Please comment on the observed differences in H3K27me3 response to Dex between ChIP-Seq (Fig. 5A) and the ChIP qPCR (Fig. 5B). Is this a timing issue?

We're guessing here, but we assume the reviewer refers to the potentially slightly higher H3K27me3 levels upon Dex treatment for ChIP-seq whereas the qPCR indicates that the levels do not change? The change seen in the ChIP-seq experiment is marginal and based on a single experiment. In contrast, the qPCR data shows the results from three biological replicates and therefore is probably a more reliable source of information indicating that H3K27me3 levels do not show a striking change upon Dex treatment.

13. Please indicate the number of replicates for the ChIP-Seq experiments in the figure legends.

We have included this information in the revised manuscript.

14. The statement "Upon hormone treatment, both the number of transcripts per cell and the number of transcriptional foci increases." on page 13 is confusing. Most cells only have two alleles (max. two transcription foci). Is *ZBTB16* duplicated in A549 cells?

Cancer cell lines very often have variable karyotypes and our FISH data suggests that the *ZBTB16* locus is present in more than 2 copies in some of the A549 cells. Here's the info from the ATCC website describing the karyotype of A549 cells: "...This is a hypotriploid human cell line with the modal chromosome number of 66, occurring in 24% of cells. Cells with 64 (22%), 65, and 67 chromosome counts also occurred at relatively high frequencies; the rate with higher ploidies was low at 0.4%....".

15. *ZBTB16* is marked by H3K27me3 (Fig. 5A/B). How many GR binding sites do overlap H3K27me3 in A549 cells? How many genes associated with GR/H3K27me3 sites are expressed in A549 cells? Is *ZBTB16* the only one?

We have looked more systematically at the link between gene regulation by GR and H3K27me3 levels to determine if this is a unique feature of the *ZBTB16* locus. In the revised manuscript, we have included a figure panel (see below, Fig. 5b) showing a H3K27me3 heatmap for genes that are upregulated upon Dex treatment. In short, as expected, the heatmap indicates that the majority of upregulated genes are not marked by H3K27me3. However, similar to *ZBTB16*, we find that a subset of genes (approx.. 10%) shows moderately high levels of this histone mark yet are not primed by prior exposure to dex. This indicates that the presence of this mark does not explain the transcriptional memory observed for the *ZBTB16* gene. The following section in the revised manuscript covers this information, page 13: “....Among genes upregulated upon hormone treatment, the presence of H3K27me3 is not unique to *ZBTB16* (Fig. 5b). Moreover, ChIP-experiments we performed showed that priming did not result in significant changes in H3K4me3 or H3K27me3 levels for either treated cells or after hormone washout indicating that these marks likely do not contribute to the transcriptional memory observed (Fig. 5c)....”

16. Is *ZBTB16* a GR target gene that is regulated by GR tissue-independently (like *GILZ* and *FKBP5*)?

We are not sure if *ZBTB16* regulation by GR is tissue independent. However, in contrast to most GR target genes that are regulated in a cell-type-specific manner, *ZBTB16* is regulated in both cell lines we examined and has also been reported to be a GR target gene in other cell types *e.g.* in macrophages (PMID: 30809020).

Reviewer #3 **Major Comments:**

1. Although the authors reported that only ZBTB16 displayed transcriptional memory, would more genes emerge with less stringent cutoffs, for example Fold Change > 1.5 & adjusted p value < 0.05?

As can be seen in figure 3D, ZBTB16 really stands out as the only gene that clearly responds more robustly upon a second hormone encounter (single red dot). Accordingly, it is the only gene that significantly changes its response upon a second hormone encounter. If we used less stringent criteria, we would eventually find additional genes that change their response, however these changes would be marginal at best and in our opinion not worth pursuing at this point.

2. One question the authors should consider is whether the washout time matters. What if it were reduced to a shorter time, for example 8 or 12 hrs? This might especially alter the conclusions about dexamethasone, which Lightman and Hager have suggested to have a long half-life of binding to the GR in cells.

For sure the washout time matters and we do not doubt that the persistent changes observed upon shorter wash-out by the Hager lab are real. One of the reasons we chose the 24h period was to see if the changes observed by Lightman and Hager might persist for extended periods of time as suggested by Zaret and Yamamoto. Our findings suggest that this is not the case and that the majority of GR-induced changes are short-lived. Perhaps future studies can shed light on how long changes persist. However, given the slow dissociation between GR and Dex, we expect that it might be hard to dissect if persistent changes are indeed persisting in the absence of GR binding or reflect an incomplete hormone wash-out.

3. The authors point out that Ref. 33 focused on persistent changes after more than 9 days, but the authors state that they focused on a 24-hour washout period which they reasoned would be more likely to reveal persistent changes. However, that was not the case, and the present findings seem to be at odds with the conclusion drawn in Ref. 33. This begs the question of whether the original report was correct and authors would have seen persistent changes (by whatever mechanism) after 9 days, or whether there almost no persistent changes at all as the present study would suggest. To address this and advance the field on this point, it is imperative that the authors do the "positive control" of repeating the protocol used in the original report, to determine if the difference is quantitative (timing) or qualitative (true discrepancy between two groups).

The objective of this study was to find out if persistent changes as observed in Ref33 are the exception or the rule, not to test if the original observation is correct (importantly, another cell line was used in Ref33 which makes a 1:1 comparison impossible to begin with). We believe that we have convincingly shown that, for the cell lines we assayed, persistent changes are rare if occurring at all. Given that no convincing persistent changes were observed after a 24h washout, we think that it is very unlikely that such changes would be observable after even longer wash-out periods. To acknowledge that other cell types may behave differently we included the following section in the discussion, page 15: "...One difference to the prior study that described persisting GR-induced hypersensitivity for more than 9 days is that they studied memory in another cell type (mouse L cell fibroblast) [38]. Another difference is that we studied genome-wide changes, whereas they studied a single exogenous stably integrated MMTV sequence. Hence, it is possible that their results do not represent a phenomenon which is commonly observed at endogenous mammalian loci. Thus, even though we did not find convincing persistence of changes in chromatin accessibility in either one of two cell lines tested, we cannot rule-out that cell type-specific mechanisms facilitate sustained accessibility...."

4. The authors state that "opening sites are often GR-occupied whereas closing sites are not occupied by GR" in Figs 1B and C. What is the fraction of opening sites with GR binding?

We agree that adding this is a good idea as this would allow for a more quantitative comparison between the different groups. We have added the following figure panels (Fig S1d for A549 and Fig S2b for U2OS-GR) that show that about 50% of the opening sites are GR-occupied whereas this number is markedly lower for closing sites.

A549 cells:

U2OS-GR cells

5. In Fig. 2C the authors show *SLC9A8* as an example of a gene which maintained a reduced level of open chromatin when assessed by ATAC-seq. To "validate" this they performed ATAC-PCR, and in the results shown in Fig. 2D any differences were not found to be statistically significant. However, these are two different assays, and both have potential flaws and experimental error. Were biological replicates of ATAC-PCR performed, and if so were the differences in ATAC-seq signal between EtOH and washout statistically significant? And is this true of other genes with similar patterns, such as *FKBP5*, *PTK2B*, and others?

For the ATAC-seq experiments, we treated the dex-treated and cort-treated experiments as replicates to find candidate regions with persistent chromatin changes. For the ATAC-seq data, a site is 'persistent' if called (by MACS2, e.g. DEX vs EtOH) upon treatment and then again 24h after washout (DEX washout vs EtOH washout). For the ATAC-qPCR experiments, we performed 4 biological replicates. In the revised manuscript, we have performed t-tests to determine if the small difference we observe at some sites between the EtOH and washout is statistically significant. In short, we did not find any statistically significant differences which we have now indicated in the ATAC-qPCR plot (Fig. 2d).

6. The authors suggest that priming increases *ZBTB16* output by increasing the fraction of cells responding to hormone treatment, but no explanation was found to explain why this happens to *ZBTB16* but not all of the other GC-induced genes. This needs to be discussed.

Indeed we did not find a mechanistic explanation for the *ZBTB16*-specific memory. Possible explanations are discussed in the following section of the results (page 15–16): "... Mirroring what we say in terms of chromatin accessibility, transcriptional responses also seem universally reversible with no indication of priming-related changes in the transcriptional response to a repeated exposure to GC for any gene with the exception of *ZBTB16*. Although several changes in the chromatin state occurred at the *ZBTB16* locus, none of these changes persisted after hormone washout arguing against a role in transcriptional memory at this locus (Fig. 5). Similarly, the increased long-range contact frequency between the *ZBTB16* promoter region and a GR-occupied enhancer does not persist after washout (Fig. 5g). Notably, our RNA FISH data showed that *ZBTB16* is only transcribed in a subset of cells, hence, it is possible that persistent epigenetic changes occurring at the *ZBTB16* locus also only occur in a small subset of cells and could thus be masked by bulk methods such as ChIP-seq or ATAC-seq. Another mechanism underlying the priming of the *ZBTB16* gene could be a persistent global decompaction of the chromatin as was shown for the *FKBP5* locus upon GR activation [35]. Likewise, sustained chromosomal rearrangements, which we may not capture by 4C-seq, could occur at the *ZBTB16* locus and affect the transcriptional response to a subsequent GC exposure. Furthermore, prolonged exposure to GCs (several days) can induce stable DNA demethylation as was shown for the tyrosine aminotransferase (*Tat*) gene [71]. The demethylation persisted for weeks after washout and after the priming, activation of the *Tat* gene was both faster and more robust when cells were exposed to GCs again [71]. Interestingly, long-term (2 weeks) exposure to GCs in trabecular meshwork cells induces demethylation of the *ZBTB16* locus raising the possibility that it may be involved in priming of the *ZBTB16* gene [72]. However, it should be noted that our treatment time (4 hours) is much shorter. Finally, enhanced *ZBTB16* activation upon a second hormone exposure might be the result of a changed protein composition in the cytoplasm following the first hormone treatment. In this scenario, increased levels of a cofactor produced in response to the first GC treatment would still be present at higher levels and facilitate a more robust activation of *ZBTB16* upon a subsequent hormone exposure. Although several studies have reported gene-specific cofactor requirements [73], the fact that we only observe priming for the *ZBTB16* gene would make this an extreme case where only a single gene is affected by changes in cofactor levels.....".

****Minor Comments****

1. What motifs are enriched at the ATAC sites that open and close?

In the revised manuscript, we have included a motif enrichment analysis in which we scanned sites which open, close or remain unchanged for the presence of JASPAR clustered motifs (see below, Fig S1c, Fig S2c). Consistent with our hypothesis that closing of sites is not driven by GR occupancy and likely indirect, we find that the GR consensus motif (cluster 15) was not enriched at closing sites.

A549 Cells:

U2OS-GR cells:

2. Fig. 1F would be improved by rephrasing the labels using terms "without site/peak" and "with site/peak". Otherwise, readers may think they are all GR peaks.

We have revised the label as suggested by the reviewer in Fig. 1d (previously Fig. 1f) and in Fig. S2d (previously Fig. S1f).

3. For Figs. 3B–3D, volcano plots are a better way to present the differentially expressed genes.

We actually prefer the MA plots as they also provide information regarding the basemean counts for regulated genes. This allows one, for example, to see that other GR-regulated genes with similar basemean counts do not show a "memory" suggesting that the low expression level for *ZBTB16* likely does not explain the observed priming.

4. p values should be shown in Figs. 6C, 6D, 6F and 6G.

We have included p-values in a revised version of the figure 6.

August 9, 2021

RE: Life Science Alliance Manuscript #LSA-2021-01080-TR

Dear Dr. Meijising,

Thank you for submitting your revised manuscript entitled "Glucocorticoid signaling induces transcriptional memory and universally reversible chromatin changes". We would be happy to publish your paper in Life Science Alliance pending final revisions necessary to meet our formatting guidelines. Please also address the Reviewer's remaining minor comments.

- please consult our manuscript preparation guidelines <https://www.life-science-alliance.org/manuscript-prep> and make sure your manuscript sections are in the correct order
- please add a conflict of interest statement to your main manuscript text
- please use capital letters when introducing panels in the actual figures, their legends, and corresponding callouts in the manuscript text
- we encourage you to revise the figure legends for Figure 3 such that the figure panels are introduced in an alphabetical order
- please add callouts for Figures 4B and S3C to your main manuscript text;

LSA now encourages authors to provide a 30-60 second video where the study is briefly explained. We will use these videos on social media to promote the published paper and the presenting author. Corresponding or first-authors are welcome to submit the video. Please submit only one video per manuscript. The video can be emailed to contact@life-science-alliance.org

A. FINAL FILES:

B. MANUSCRIPT ORGANIZATION AND FORMATTING:

Sincerely,

Reviewer #2 (Comments to the Authors (Required)):

The authors provide a resourceful analysis of GC-induced alteration in histone modifications and DNA accessibility as well as mRNA expression upon recurring GC treatment. The two analysed cell lines (A549, U2O2-GR) do not show maintained memory of GR exposure on gene expression (priming for 4h) or DNA accessibility (priming for 24h). The authors however, also observe some cell-type specificity. U2O2-GR cells, but not A549 cells, maintained GR binding and "openness" as well as their H3K27ac at the GR enhancers even after 24h washout. Those maintained GR binding is explainable by remaining low-level GCs in the media or the fact that U2O2-GR cells overexpress the GR. A459 cells but not U2O2 on the other hand showed memory effect at a single gene after repeated GC exposure. ZBTB16 was accumulative induced by repetitive GC exposure (with 24h hormone withdrawal in-between). The authors convincingly show that the observed "priming" of the ZBTB16 gene is an effect of population-based analysis. Single cell FISH experiments revealed that subsequent GC exposures induce a ZBTB16 expression in an increasing fraction of cells. Bulk analysis of 3D genome architecture or histone modifications at this locus did not yield a mechanistic explanation, but might be hampered by their signal accumulation across the cell population.

The study presented by Bothe et al. convincingly indicates that most of the GC effects are reversible (gene expression, DNA accessibility, histone modifications), which is compatible with the circadian nature of GC release in mammals. This study will also provoke and initiate further investigations into a GC-driven cellular or transcriptional memory as contradicting observations were made in other cell types and the authors themselves observe cell type specific memory on ZBTB16. Furthermore, highly relevant questions remain concerning the exposure time to GCs as many clinical treatment schemes for example disrupt the "natural" GC cycle and do not allow for 24 washout periods. However, those questions as well as more detailed in-vivo analysis are beyond the scope of this study.

The manuscript itself is well-structured and written, the methods are extensively explained and the data well presented.

The authors thoroughly addressed most of the previous comments. I have some minor comments.

1. At the end of the introduction (page 4), the authors state the "cells may remember a previous exposure to hormone in a gene-specific manner". As they only observe memory in A459 cells, I suggest to additionally include in "gene-specific and cell type-specific manner".
2. In the last paragraph of the "Linking GR occupancy, chromatin accessibility and gene regulation" section on page 10, the authors state "Instead, our data are consistent a 'squenching model' whereby repression is driven by a redistribution of cofactors...". This conclusion cannot be made from the presented data. The authors, for example, show an enrichment of AP-1 motifs in "closing" sites in A459 and U2O2-GR cells. GC-induced changes in gene expression or activation of AP-1 isoforms might be another mechanism (among many).
3. The authors describe that ZBTB16 is activated in an increasing subsets of A459 cells after repeated GC exposure by scFISH, which is part of the memory mechanism. As U2O2-GR cells did not show a memory for ZBTB16, it would be interesting to investigate the fraction of U2O2-GR cells expressing ZBTB16 after the first GC treatment.

Dear Dr. Sawey and dear reviewers of our manuscript,

We're excited to see our work published in LSA. In a revised version of the text and figures, we have implemented the requested revisions to meet Life Science Alliance's formatting guidelines. In addition, we have made one textual change in response to reviewer #2's remaining minor comments.

Below please find attached our responses to the reviewer's comments.

Sincerely,

Sebastiaan H. Meijzing

Reviewer #2: **Minor comments**

1. At the end of the introduction (page 4), the authors state the "cells may remember a previous exposure to hormone in a gene-specific manner". As they only observe memory in A459 cells, I suggest to additionally include in "gene-specific and cell type-specific manner".

We have incorporated the textual change as suggested (page 4).

2. In the last paragraph of the "Linking GR occupancy, chromatin accessibility and gene regulation" section on page 10, the authors state "Instead, our data are consistent a 'squelching model' whereby repression is driven by a redistribution of cofactors...". This conclusion cannot be made from the presented data. The authors, for example, show an enrichment of AP-1 motifs in "closing" sites in A459 and U2O2-GR cells. GC-induced changes in gene expression or activation of AP-1 isoforms might be another mechanism (among many).

We agree that our results do not "prove" that repression is driven by a squelching model. However, we stand by our statement that our findings (*e.g.* a re-distributions of p300 away from closing sites not occupied by GR to GR-occupied sites) are consistent with squelching-driven repression. This does not exclude other possible mechanisms, which is acknowledged in the following section, page 5: "...*Instead, our data are consistent a 'squelching model' whereby repression is driven by a redistribution of cofactors away from enhancers near repressed genes that become less accessible upon GC treatment yet lack GR occupancy. However, repression may also be driven by other mechanisms....*"

3. The authors describe that ZBTBT16 is activated in an increasing subsets of A459 cells after repeated GC exposure by scFISH, which is part of the memory mechanism. As U2O2-GR cells did not show a memory for ZBTB16, it would be interesting to investigate the fraction of U2O2-GR cells expressing ZBTB16 after the first GC treatment.

This would indeed be interesting, but unfortunately not something that we can easily add at the moment.

August 12, 2021

RE: Life Science Alliance Manuscript #LSA-2021-01080-TRR

Dr. Sebastiaan H. Meijsing
Max Planck Institute for Molecular Genetics, Berlin, Germany
Computational Molecular Biology
Innestrasse 63-73
Berlin 14195
Germany

Dear Dr. Meijsing,

Thank you for submitting your Research Article entitled "Glucocorticoid signaling induces transcriptional memory and universally reversible chromatin changes". It is a pleasure to let you know that your manuscript is now accepted for publication in Life Science Alliance. Congratulations on this interesting work.

DISTRIBUTION OF MATERIALS:

Again, congratulations on a very nice paper. I hope you found the review process to be constructive and are pleased with how the manuscript was handled editorially. We look forward to future exciting submissions from your lab.

Sincerely,
